

# Impact of precipitation and increasing temperatures on drought in eastern Africa

Sarah F. Kew[1], Sjoukje Y. Philip[1], Mathias Hauser[2], Mike Hobbins[3,4], Niko Wanders[5], Geert Jan van Oldenborgh[6], Karin van der Wiel[6], Ted I.E. Veldkamp[1], Joyce Kimutai[7], Chris Funk[8,9], and Friederike E.L. Otto[10]

[1]Institute for Environmental Studies, Vrije Universiteit, Amsterdam, The Netherlands
[2]Institute for Atmospheric and Climate Science, ETH Zurich, Zurich, Switzerland
[3]Cooperative Institute for Research in Environmental Sciences, University of Colorado Boulder, Boulder, Colorado
[4]Physical Sciences Division, NOAA/Earth System Research Laboratory, Boulder, Colorado
[5]Department of Physical Geography, Utrecht University, Utrecht, the Netherlands
[6]Royal Netherlands Meteorological Institute (KNMI), De Bilt, The Netherlands
[7]Kenya Meteorological Department, Nairobi, Kenya
[8]U.S. Geological Survey Center for Earth Resources Observation and Science, Sioux Falls, South Dakota
[9]University of California, Santa Barbara, Santa Barbara, California
[10]School of Geography and the Environment, University of Oxford, Oxford, UK

**Correspondence:** Sarah F. Kew (sarah.teulingkew@gmail.com)

**Abstract.** In eastern Africa droughts can cause crop failure and lead to food insecurity. With increasing temperatures, there is an a priori assumption that droughts are becoming more severe, however, the link between droughts and climate change is not sufficiently understood. In the current study we focus on agricultural drought and the influence of high temperatures and precipitation deficits on this.

Using a combination of models and observational datasets, we studied trends in six regions in eastern Africa in four drought-related annually averaged variables — soil moisture, precipitation, temperature and, as a measure of evaporative demand, potential evapotranspiration (PET). In standardized soil moisture data, we find no discernible trends. Precipitation was found to have a stronger influence on soil moisture variability than temperature or PET, especially in the drier, or water-limited, study regions. The error margins on precipitation-trend estimates are however large and no clear trend is evident. We find

significant positive trends in local temperatures. However, the influence of these on soil moisture annual trends appears limited as evaporation is water limited. The trends in PET are predominantly positive, but we do not find strong relations between PET and soil moisture trends. Nevertheless, the PET-trend results can still be of interest for irrigation purposes as it is PET that determines the maximum evaporation rate.

We conclude that, until now, the impact of increasing local temperatures on agricultural drought in eastern Africa is limited

and recommend that any soil moisture analysis be supplemented by analysis of precipitation deficit.



## 1 Introduction

In eastern Africa, drought has occurred throughout known history, and the phenomenon has incurred significant impacts on the agricultural sector, and hence on the regional population and economy, particularly thorough threats to food security. It is therefore important to examine the role of anthropogenic climate change in drought, particularly in the face of the large-scale droughts of 2010/11, 2014, and 2015 in Ethiopia, and the 2016/17 drought in Somalia, Kenya, parts of Ethiopia and surrounding countries, which have recently raised the spectre of climate change as a risk multiplier in the region.

The complexity of droughts poses particular challenges for their attribution. Droughts are triggered and maintained by a number of factors and their interactions, including meteorological forcings and variability, soil and vegetation feedbacks, and human factors such as agricultural practices and management choices, including irrigation and grazing density (van Loon et al., 2016). There are four main definitions of drought (Wilhite and Glantz, 1985): meteorological drought (precipitation deficit), hydrological drought (low streamflow), agricultural drought (low soil moisture), and socioeconomic drought (including supply and demand). In the current study we focus on agricultural drought and how soil moisture and evapotranspiration is influenced by precipitation deficits and high temperatures.

With respect to meteorological drought drivers, the link between drought and climate change is not sufficiently understood. Some weather stations in eastern Africa have recorded a decrease in precipitation in recent years as regional temperatures have increased in line with the global mean. While the regional temperature increase can be attributed to anthropogenic climate change, the effect of a changing climate on precipitation is generally much less straightforward (Hauser et al., 2017), particularly in eastern Africa. Climate models generally project an increase in mean precipitation but give conflicting results for the probability of very dry rainy seasons (e.g. Shongwe et al., 2011). Recent studies have found little or no change in the risk of low-precipitation periods due to anthropogenic climate change (e.g., Philip et al., 2018a; Uhe et al., 2018). The reasons for the recent observed decrease in precipitation thus remains unclear, but the trend is within the large observed natural variability in the region, at least for the historical and current climate. However, precipitation only covers one aspect of drought — that of the supply side of the water balance. The demand side is represented by actual evapotranspiration (ET), which is a function of moisture availability and evaporative demand (also referred to as "potential evapotranspiration" PET). PET is itself a function of temperature, humidity, solar radiation, and wind speed. With increasing temperatures, there is an a priori assumption that rising PET will increase the demand side of the water balance and, all else equal, droughts will become more severe. However, this assumption is not based on analyses, which motivates an objective study.

Aside from meteorological drought studies, the majority of drought studies on eastern Africa have focused on hydrological droughts. Krysanova et al. (2017) analysed high and low river flows including those on the Blue Nile. They found no significant change in low flows, but point out that uncertainties are large for low flows due to uncertainties inherent in Global Climate Models (GCMs), climate scenarios, and hydrological models. This was confirmed by Huang et al. (2017), who identify no clear trends in low flows in the Blue Nile basin. Prudhomme et al. (2014) studied low flows on the global scale using a combination of models, comparing 2070–2099 to 1976–2005, and found little change in or reduced occurrence of drought projected across the Horn of Africa.





Across eastern Africa, agricultural drought is intimately linked to the quality and quantity of food production for domestic consumption. In our analysis we focus on annual drought, as the worst crises in food security in this region have occurred with multiple season droughts (Funk et al., 2015). Drivers of food security include precipitation and precipitation-related factors like reduced food production, low livestock prices, high food prices, and reduction in food access. Non-climatic factors are

also important; these include population change, land-use change and change in water demand. Pricope et al. (2013), and Coughlan de Perez et al. (2019) show that vulnerability has a large influence on food security, e.g., socio-economic factors, stability, and the ability to trade with other regions. Food-security outcomes differ dramatically between different regions and between pastoralist and non-pastoralist livelihood zones: pastoral and agro-pastoral populations are more often food insecure than non-pastoral populations. Not only are these non-climate-related factors often more significant than climate (change) in

determining food security, but they are often unpredictable. Thus, attributing crop growth to climate change — i.e., isolating the influence of climate change on crop growth — can be challenging and does not tell the full story. Hansen et al. (2016) argue that, if a good observational basis of factors influencing food security exists, the detection and attribution of impact-related variables to climate change is possible. If not, it is more informative to perform the climate change attribution on a variable less directly related to food security but for which the best observational basis exists. In this paper we aim for an intermediate

step: the choice of variables is a compromise between the best-observed variable of relevance (precipitation) and the variables best related to food security/crop growth.

We study agricultural drought – defined as low soil moisture – as soil moisture is a better indicator of crop health than precipitation and is an important indicator in regions without irrigation. We discuss the influence of meteorological drivers on soil moisture, i.e., precipitation and temperature. Ideally, we would study the influence of temperature on soil moisture via

evapotranspiration (ET), however observational records are very limited in time and space and as the spatial correlation lengths of evapotranspiration are short their informational value is limited. To more directly explore the influence of temperature on soil moisture we therefore analyse evaporative demand, choosing as a measure PET (the amount of evaporation that would occur if an unlimited supply of water were available), which is calculable/available for both observations and model simulations. Hydrological models driven by GCM or reanalysis data usually either take PET directly as an input or derive PET from the

different components required by the chosen PET scheme. The theoretically maximum evaporation, as initially given by PET, is then limited by available soil moisture to simulate actual ET. Evaporative demand can be regarded as even more important than soil moisture for regions with irrigation or where irrigation is considered. Therefore, a step between attribution of food security and attribution of precipitation is the attribution of soil moisture and additionally PET.

PET has been used in various attribution studies outside our region of study, to explore specific events or causes of trends

or changes in variability. For China, PET was used to study the influence of climate change on the hydrological cycle (e.g. Yin et al., 2010; Li et al., 2014; Fan and Thomas, 2018). Obada et al. (2017) study trends and variability in PET at different sites in West Africa, showing high variability of PET when using Penman-Monteith PET and station data for six sites in Benin. Manning et al. (2018) used sites in Europe to study compound events of low precipitation and high PET. One of their findings is that while precipitation has the most influence on soil moisture for both wet and dry regions, the influence of PET

on soil moisture differs between regions: in wet regions, high values of PET, signifying atmospheric conditions conducive to





evaporation, can amplify low soil moisture anomalies during drought, whereas in dry regions where very little surface moisture is available for evaporation, the magnitude of PET has little influence on soil moisture. Coughlan de Perez et al. (2019) choose a drought index that does not include PET when assessing the utility of precipitation observations to anticipate food insecurity in eastern Africa, as including PET made little difference to the results and most observational data (e.g. temperature data)

required to calculate PET is likely to be less accurate than observed precipitation data in this region.

Dewes et al. (2017) examined the efficacy of evaporative demand ($E_0$) in assessing drought projections across the generally water-limited US Northern Great Plains, comparing the variability in projected drought risk from temperature-, radiation- and Penman-Monteith-based $E_0$ parameterizations, two drought indices, and two CMIP5 climate driver sets. They found much larger $E_0$ trends in temperature- and radiation-based $E_0$ estimates, and a greater uncertainty due to the choice between these

than to the choice of GCM drivers. They also found significant difference in projected drought risk as estimated by $E_0$-based and a combined precipitation-$E_0$ index (i.e., the SPEI). Within eastern Africa, Hobbins et al. (2016) decomposed the variability of $E_0$ (using the Penman-Monteith reference ET dataset used in this study), and found that the dominant driver of daily $E_0$ variability varied across time and space, with all four drivers (temperature, solar radiation, humidity, and wind speed) variously dominating across the continent, but particularly across eastern Africa, where the spatial picture strongly reflected

the heterogeneous topography and hydroclimate and the external sources of variation in the drivers.

The objectives of this study are to (i) consider the question "do increasing global temperatures contribute to drier soils and thus exacerbate the risk of agricultural drought (low soil moisture)?" and (ii) more fully understand the interplay between temperature and precipitation and their influence on agricultural drought. Assessments will be based on both observations and climate and hydrological models. We will relate the findings to examples of recent droughts in eastern Africa.

In the following sections we first describe the different study regions in eastern Africa and the datasets used to provide the four different variables we analyse — soil moisture, precipitation, temperature, and PET. We include brief descriptions of the (modelling) projects from which the datasets originate. In Section 3 we describe the methods used in this paper, including validation of the data and calculation of trends. Sections 3.1 and 3.2 list the assumptions and decisions and an example of the method, respectively. Next, the results are synthesized per region. Finally, a discussion and conclusions are presented in

Sections 5 and 6.

## 2   Study region and datasets

In this section we show the regions under analysis, list the datasets used and summarize their advantages and drawbacks.

### 2.1   Study region

Trend analyses of time series of regionally averaged quantities are more robust than the same analyses for point locations.

This is especially true for precipitation, which shows small-scale spatial variability if the time period is not long enough to sufficiently sample the distribution from multiple precipitation events. It is however necessary to select homogeneous zones, so that the signals present are not averaged out.



**Table 1.** The six study regions. See also Fig. 1

| Region | Long name | Latitude | Longitude | Primary land-use type |
|--------|-----------|----------|-----------|----------------------|
| WE | West Ethiopia | 7°N-14°N | 34°E-38°E | agropastoral/mixed land |
| EE | East Ethiopia | 8°N-13°N | 38°E-43°E | pastoral |
| NS | North Somalia/Somaliland region and East Ethiopia | 5°N-12°N | 43°E-52°E | pastoral |
| NK | North Kenya | 2°N-4.5°N | 34°E-41°E | pastoral |
| CK | Central Kenya | 1.5°S-1.5°N | 35°E-38.5°E | agropastoral/mixed land |
| SS | South Somalia | 2°S-5°N | 41°E-48°E | pastoral/agropastoral |

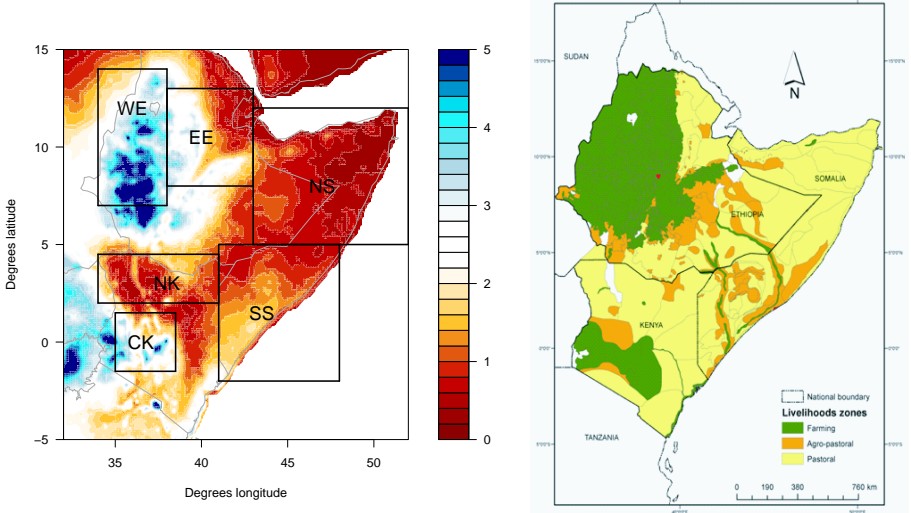

**Figure 1.** Left: annual mean precipitation [mm/day] and the six study regions. Note that only land values are used. Right: livelihood zones after Pricope et al. (2013), which were also used to define the study regions.

The focus of the study is on eastern Africa — Ethiopia, Kenya, and Somalia (including the Somaliland region). We selected six regions based on livelihood zones, homogeneous precipitation zones and local expert judgement. The regions are shown in Fig. 1 and listed in Table 1.

## 2.2 Datasets

5 We analyse four different variables: soil moisture, precipitation, temperature, and PET. Soil moisture and PET are estimated using data-model chains. For the four variables, we use as many datasets as readily available, provided that the data are sufficiently complete over a long-enough time period to be used for trend calculations. For this purpose, we decided to use time series of 35 years and longer. As the focus of this paper is on annual time scales, using monthly data is sufficient. The datasets



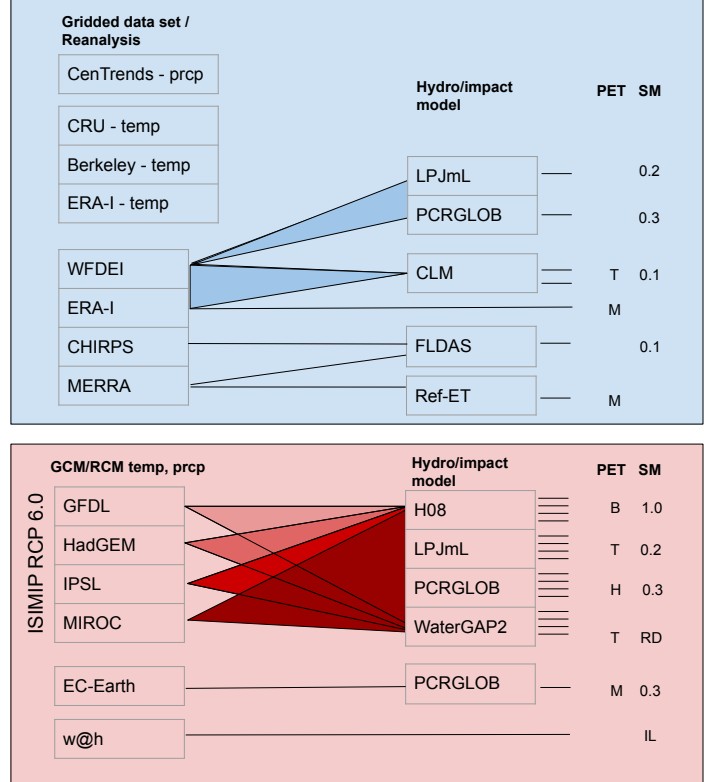

**Figure 2.** Datasets used in this paper. Top: observational precipitation (prcp) and near-surface temperature (temp) datasets, bottom: models. Listed under PET is the PET scheme (T: Priestley-Taylor, M: Penman-Monteith, H: Hamon, B: Bulk formula) and, under SM, is the depth of the top soil moisture layer available (RD: depends on rooting depth (0.1-1.5m for WaterGAP2); IL: integrated over all layers). Shading indicates an experiment with either multiple input datasets or multiple hydrological models. The number of resulting hydrological model simulations are indicated by horizontal lines on the right side of the figure.

used in this study are shown in Fig. 2 and listed below. Brief project and (model) data descriptions are given below the list. Note that we use the data as is available without applying any additional bias correction. Some of the data has undergone bias correction in the projects, as listed below.

For observations and reanalyses we use (see below for more detailed explanations):

5   –   For soil moisture: CLM-ERA-I, CLM-WFDEI, FLDAS (McNally et al., 2017) and because soil moisture is only calcu-
lated with models we additionally used the two available ISIMIP runs with WFDEI reanalysis input (PCRGLOB-WFDEI
Sutanudjaja et al. (2018) and LPJmL-WFDEI).





- For PET: CLM-ERA-I, CLM-WFDEI, MERRA RefET (Hobbins et al., 2018), ERA-I (Dee et al., 2011)

- For precipitation: CenTrends (Funk et al., 2015)

- For near-surface temperature: CRU TS4 (Harris et al., 2014), Berkeley (anomalies), ERA-Interim (Dee et al., 2011)

For model data the following simulations are used:

- 16 (4 GCMs x 4 hydrological models) transient runs from the ISIMIP ensemble that have data available for both soil moisture and PET.

- daily mean near-surface temperature and precipitation series from the 4 GCMs that are used in these ISIMIP runs.

- The combination EC-Earth - PCR-GLOBWB for all four variables

- The weather@home ensemble for near-surface temperature, precipitation and soil moisture

The reanalysis and model data are obtained from different model projects. A brief description of these projects and models is given below, see the references herein for more details.

The Community Land Model version 4 (CLM, Lawrence et al. (2011); Oleson et al. (2010)) was forced with 6-hourly atmospheric forcing from two datasets. Once with raw ERA-Interim data (CLM-ERA-I), and once with bias-corrected ERA-Interim data from the Water and Global Change (WATCH) project; i.e., the WATCH-Forcing-Data-ERA-Interim (WFDEI) (Weedon et al., 2014) (CLM-WFDEI). CLM is the land component of the Community Earth System Model and has 10 hydrologically active soil layers with exponentially increasing thickness, as well as a groundwater module.

The daily reference ET (RefET; Hobbins et al. (2018)) used in this study was generated by the US National Oceanic and Atmospheric Administration from hourly drivers provided by the Modern-Era Retrospective analysis for Research and Applications, Version 2 (MERRA-2) of the US National Aeronautics and Space Administration (NASA) Global Modeling and Assimilation Office (GMAO). Hourly MERRA-2 drivers for 2-m air temperature, 2-m specific humidity, surface pressure, 2-m wind speeds, and downwelling shortwave radiation at the surface were aggregated to daily input to the American Society of Civil Engineers Standardized Reference ET Equation [Allen et al. (2005); identical to the international standard FAO-56 report (Allen et al., 1998) at the daily timescale] for a 0.12-m short grass reference crop. The reference ET data are available daily from January 1, 1980 to within a few weeks of the present at a resolution of 0.125°, which was downscaled from the native MERRA2 resolution of 0.5°latitude × 0.625°longitude (https://www.esrl.noaa.gov/psd/eddi/globalrefet/).

Within the second phase of the Inter-Sectoral Impact Model Intercomparison project (ISIMIP2 Gosling et al., 2019), global hydrological models are used for intercomparison of climate impacts. The model simulations use input from GCMs or observations, e.g., precipitation and PET. For more details on the ISIMIP project, please refer to Warszawski et al. (2014) and Rosenzweig et al. (2017). We used modelled soil moisture (0.5° × 0.5°spatial resolution) for the period 1850/1861–2018 from four global hydrological models: H08 (Hanasaki et al., 2008a, b), LPJmL (Bondeau et al., 2007; Rost et al., 2008; Schaphoff et al., 2013), PCR-GLOBWB (van Beek et al., 2011; Wada et al., 2011, 2014) and WaterGAP2 (Müller Schmied et al., 2016). The GCMs used are GFDL-ESM2M (GFDL, Dunne et al., 2012, 2013), HadGEM2-ES (HadGEM, Jones et al., 2011), MIROC5



(MIROC, Watanabe et al., 2010), IPSL-CM5A-LR (IPSL, Dufresne et al., 2013). All simulations were carried out under the modelling framework of phase 2b of the Inter-Sectoral Impact Model Intercomparison Project (ISIMIP2b: Frieler et al., 2017, https://www.isimip.org/protocol/#isimip2b). For the ISIMIP models we also have access to the adjusted (i.e., bias corrected for the ISIMIP project) GCM data that is used as input for the hydrological models (see also Section 3.1). For precipitation and
temperature, the original GCM data is analysed for trends.

For weather@home we use the large-ensemble regional modelling approach as in (Uhe et al., 2018) employing the distributed computing framework climateprediction.net (Massey et al., 2015; Guillod et al., 2017). For this study, two large ensembles of simulations of temperature, precipitation and soil-moisture are available for the present day climate (2005–2016) and a counterfactual climate representing how conditions might have been without anthropogenic greenhouse gas and aerosol emissions
for the same time-frame.

EC-Earth - PCR-GLOBWB has been developed using large-ensemble simulations of EC-Earth (Hazeleger et al., 2012) in combination with the PCR-GLOBWB model (Sutanudjaja et al., 2018). The EC-Earth - PCR-GLOBWB (EC-PCRGLOB) ensemble was originally developed by van der Wiel et al. (2019) to study changes in hydrological extremes.

There is little available observational data for soil moisture, with most series being too short to use for trend analysis.
McNally et al. (2016) used CCI satellite-derived microwave soil moisture data to show that observational, reanalysis and model datasets for soil moisture are not always highly correlated with each other over eastern Africa. Using a set of models and observations or assimilated data for soil moisture is therefore always necessary to span the large uncertainties from inter-dataset differences. There being no a priori reason to favour one soil moisture dataset over another, we treated all resulting soil moisture datasets equally. For soil moisture we use the topmost layer provided by each dataset and scale each time series to
have a standard deviation of 1 in order to make comparisons in trends possible. An exception to this is weather@home where the available soil moisture variable is an integrated measure of all four layers of soil moisture in the model, including the deep soil.

Wartenburger et al. (2018) show that the influence of different choices in the calculation of PET and soil moisture can be quite large, with two of the choices being the input dataset for the hydrological model and PET scheme. Trambauer et al.
(2014) also showed that both the scheme used to calculate PET and the input data used for calculation of PET have a large influence on PET values. We confirm this using the CLM-ERA-PT (Priestley-Taylor), CLM-WFDEI-PT and CLM-ERA-PM (Penman-Monteith) datasets (not shown). In our study regions, PET values are consistently higher when using PM then when using PT. The differences in trends in PET using ERA or WFDEI input or using PT or PM input are sometimes significant. However, comparing study regions, there is no consistency in the difference; in four out of the six regions the PM data shows a
higher trend than the PT data and in four out of the six regions WFDEI data shows a higher trend than the ERA data. We thus chose to use a variety of PET parameterizations and input datasets in order to cover the range of possible PET values and trends in PET. If several schemes are available there is an a priori preference for a Penman-Monteith PET scheme, as demonstrated by Hobbins et al. (2016), who showed that the dominant driver of RefET variability is often neither temperature nor radiation. However, one is often constrained from using a Penman-Monteith parameterization due either to the lack of accurate or reliable



input data or because the choice of PET parameterization within a given hydrological model setting is already prescribed, as in the ISIMIP ensemble.

It is still unknown how vegetation will respond to substantial increases of $CO_2$ concentration. Two counteracting effects — physiological and structural responses — are expected, but their net effect is unknown (e.g. Wada et al., 2013). The phys-
iological response is that increased $CO_2$ results in increased water use efficiency (WUE), that is reduced water loss through evapotranspiration by reducing stomatal openings. The structural response is increased growth, which may lead to increased leaf area and thereby increased water loss through evapotranspiration (Prudhomme et al., 2014). So-called 'dynamic vegetation models' include these $CO_2$ effects, whereas others do not. Wada et al. (2013) studied uncertainties of irrigation water demand under climate change and found much lower demand is simulated by global impact models that model the $CO_2$ effect than
those that do not. Prudhomme et al. (2014) found that models that include the $CO_2$ and vegetation effects show a weaker response of global hydrological drought to climate change. They therefore advise users to ensure that the impact (hydrological) models they select must capture the diverse ways that models represent the response of plants to $CO_2$ and climate — the largest source of uncertainty in their study. In this study we use different combinations of observational input datasets, GCM input and hydrological models. Our selection of hydrological models is restricted by the variables we require, however, out of the four
ISIMIP hydrological models that match our criteria, one (LPJmL) uses dynamic vegetation modeling.

With the datasets we use we cover a wide range of different factors that influence PET and soil moisture. The different factors include meteorological forcing, model choice, RCP scenario for the greenhouse gas concentration trajectory, PET scheme, number of soil layers and depth of topsoil layer, and transient versus time slice runs (see next section on 'Methods').

## 3  Methods

In this section we describe first the method we use for detection and attribution of trends in the four variables, and then the validation of the data. Finally we outline the synthesis method.

We use a multi-method, multi-model approach to address attribution. We calculate trends with respect to global mean surface temperature (GMST) for all variables, regions and datasets and synthesize results into one overarching attribution statement for each of the four variables in each of the six regions. We use this method, following the approach applied in earlier studies
on drought in eastern Africa (e.g., Philip et al., 2018a; Uhe et al., 2018) and other drought- and heat-attribution studies (e.g., Philip et al., 2018b; van Oldenborgh et al., 2018; Kew et al., 2019; Sippel et al., 2016), which represents the current state of the art in extreme event attribution. The method is extensively explained in van Oldenborgh et al. (2019) and Philip et al. (2019).

In this study, for transient model runs and observational time series, we statistically model (i.e., fit) the dependency of annual means of the different variables on GMST, (the model GMST for models, and GISTEMP surface temperature GMST (Hansen
et al., 2010) for observations and reanalyses) as follows (see also van Oldenborgh et al., 2019; Philip et al., 2019):

We use the following distributions:

- for soil moisture: a Gaussian distribution that scales with GMST, focussing on low values,

- for precipitation: a General Pareto Distribution (GPD) that scales to GMST, analyzing low extremes





- for temperature: a Gaussian distribution that shifts with GMST, focussing on high values, and

- for PET: a Gaussian distribution that scales with GMST, focussing on high values.

When the distribution is shifted, a linear trend $\alpha$ is fitted by making the location parameter $\mu$ dependent on GMST as

$$\mu = \mu_0 + \alpha T, \tag{1}$$

with $\alpha$ in [units of the study variable]/K. When the distribution is scaled,

$$\mu = \mu_0 \exp(\alpha T/\mu_0), \tag{2}$$
$$\sigma = \sigma_0 \exp(\alpha T/\mu_0), \tag{3}$$

which keeps the ratio of the location and scale parameter $\sigma/\mu$ invariant. In each case, the fitted distribution is evaluated twice: once for the year 1900 and once for the year 2018. To obtain a first-order approximation of the percentage change between

the two reference years, $\alpha$ is multiplied by 100% times the change in GMST and divided by $\mu_0$ (for the shift fit this is exact). Note that for some variables — e.g., precipitation — it is appropriate to scale rather than shift the distribution with GMST (see van Oldenborgh et al., 2019; Philip et al., 2019, for an explanation). For the very large weather@home ensemble simulations of actual and counterfactual climates, it is not necessary to use a fitting routine as the large amount of data permits a direct estimation of the trend. This also provides an opportunity to check the assumptions made in the fitting, notably that the values

follow an extreme-value distribution and that the distribution shifts or scales with the smoothed GMST. We calculate trends for the time series of spatially and annually averaged data of all four variables and all six regions for all datasets by dividing the difference in the variable between the two ensembles by the difference in GMST.

Figs. 3 and 4 present the methods applied to transient series and time slices respectively. For reference and to aid interpretation of the return-period diagrams, the magnitude of a hypothetical event with a 20-year return period in the year 2018 or in

the current climate is shown as a horizontal line or square. Reading the return period at which this line crosses the fit for the reference year 1900 shows how frequent an event with a 20-year return period in today's climate would have been then.

We only use results from model runs if they pass two different validation tests — a qualitative test on the seasonal cycle and a stronger test on variability. For soil moisture, due to the difficulties in obtaining reliable soil moisture measurements (e.g., Liu and Mishra, 2017) and the differences between the observational (reanalysis) datasets, we cannot assume that observational or

reanalysis data are more accurate than model data. Therefore we simply use the soil moisture model data if the model input — PET and precipitation — passed the validation tests.

We perform only a qualitative validation of the seasonal cycle. For each region, each variable, and each model we check that the seasonal cycle resembles that of at least one of the observational datasets, in both the number and the timing of peaks. If the seasonal cycle is very different, we do not use the time series for that specific combination. This is the case for the original

GCM precipitation in region NK for weather@home and in regions NK and CK for MIROC (the seasonal cycle is improved in the adjusted dataset, so we still use the time series in soil moisture) and for temperature in region SS for EC-Earth (we do not have adjusted data to check so we do not use this model-region combination for soil moisture or PET).



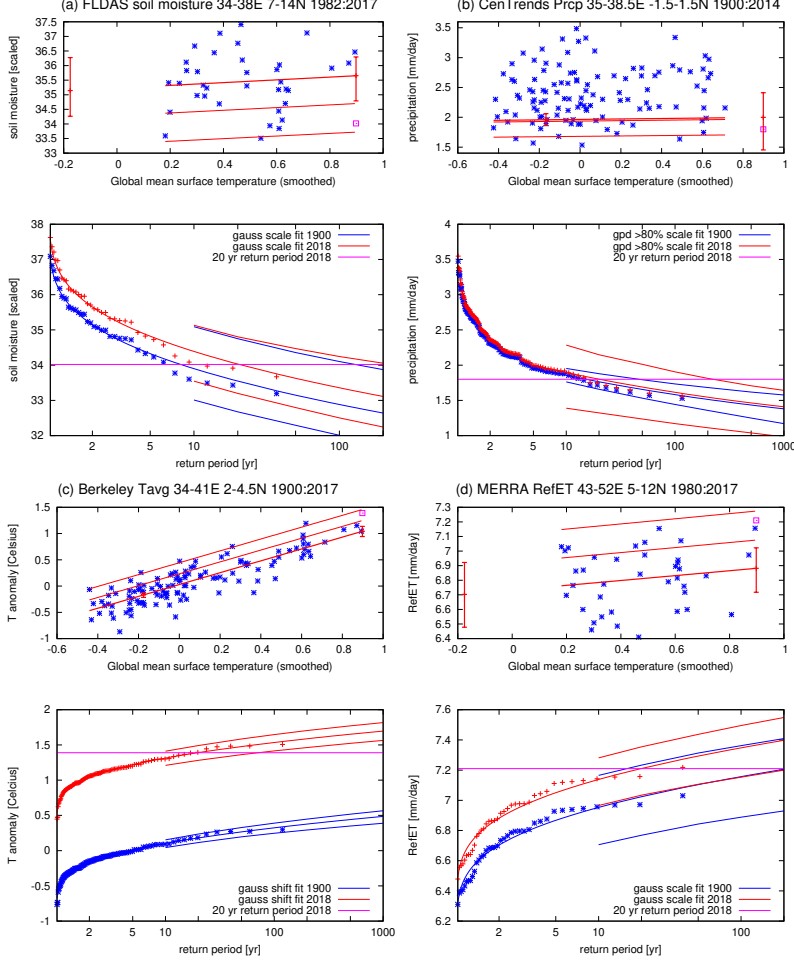

**Figure 3.** Illustrative examples of the fitting method for each variable, for selected study regions. (a) FLDAS soil moisture (Gauss fit, low extremes, region WE); (b) CenTrends precipitation (GPD fit, low extremes, region CK); (c) Berkeley temperature anomaly (Gauss fit, high extremes, region NK); (d) MERRA PET (Gauss fit, high extremes, region NS). Top of each panel: annually averaged data (stars) against GMST and fit lines - the location parameter $\mu$ (thick), $\mu \pm \sigma$ and $\mu \pm 2\sigma$ (thin lines, Gaussian fits), and the 6 and 40 year return values (thin lines, GPD fit). Vertical bars indicate the 95% confidence interval on the location parameter $\mu$ at the two reference years 2018 and 1900. The magenta square illustrates the magnitude of an event constructed to have a 20-year return period in 2018 (not included in the fit). Bottom of each panel: return period diagrams for the fitted distribution and 95% confidence intervals, for reference years 2018 (red lines) and 1900 (blue lines). The annually averaged data is plotted twice, shifted or scaled with smoothed global mean temperature up to 2018 and down to 1900. The magenta line illustrates the magnitude of a hypothetical event with a 20-year return period in 2018.

The second validation test is on the model variability in precipitation and PET (variability relative to the mean for variables that scale with GMST). If the model variability of a specific variable in a specific region is outside the range of variability calculated from observations or reanalyses, we do not use that specific dataset for that specific region and variable. For tem-



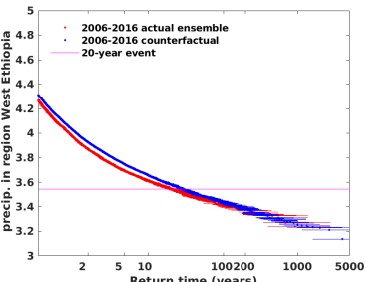 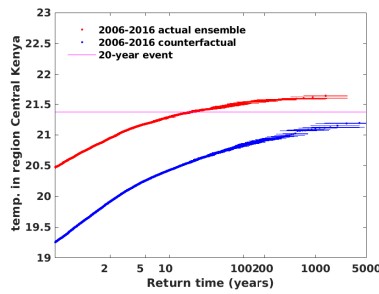

**Figure 4.** Illustrative examples of the weather@home time slice model runs. Left: annual mean precipitation in region WK. Right: annual mean temperature in region CK. The red markers are for the present day climate and the blue markers are for the climate in pre-industrial times. The magenta line illustrates the magnitude of a hypothetical event with a 20-year return period in the present day climate.

perature, we relax the validation criteria on variability as during the analysis it became clear that the trend in soil moisture does not depend strongly on temperature and the trend in temperature agrees between models and observations. In two of the regions a strict validation resulted in only two driving GCMs. Trends from the resulting time series that passed the validation tests are shown in Section 4 and in the figures in the Supplementary Information.

Using the large weather@home ensemble (which requires no fitting), we check the assumption that precipitation and soil moisture scale with GMST and temperature shifts with GMST. For precipitation, we assume that the distribution scales with GMST. In the weather@home ensemble, dry extremes show less change than intermediate dry extremes, which supports our assumption that scaling with GMST is appropriate (except for the higher return values, where the uncertainties are large). For soil moisture it is very difficult to distinguish between scaling and shifting from the weather@home ensemble because the trend is small. For temperature the weather@home ensembles indicate that the highest temperatures are increasing slower than the lower temperatures. This implies that the variability decreases with GMST, however no consistent signal in the observations or other models is evident (we see a small increase in variability with time for Berkeley, a small decrease for CRU, and no consistency between the models). This does not affect the trend much, which is evaluated for the centre of the distribution.

Trends are presented as change in a variable per degree of GMST warming. We show trends rather than probability ratios, which conveniently results in finite ranges in confidence intervals for all variables. This is not the case for the probability ratio, where, for example, strong trends in temperature imply that mild extremes of the 2018 climate (e.g., a 1-in-20-year event) would have had a chance of almost zero around 1900, resulting in very large probability ratios and extensive extrapolation of the fit beyond the length of the dataset.

We synthesize the trends of all data that passes the validation tests in the following manner, see also Fig. 5. The observational (reanalysis) estimates are all based on the same natural variability, the historical weather. They also cover similar time periods. The uncertainties due to natural variability (denoted as solid blue in the synthesis figures in the synthesis figures) are therefore highly correlated. We approximate these correlations by assuming the natural variability to be completely correlated, and compute the mean and uncertainties as the average of the different observational estimates. The spread of the estimates is



a measure of the representation uncertainty in the observational estimate and is added as an independent uncertainty to the natural variability (black outline boxes). This results in a consolidated value for the observations (reanalyses) drawn in dark blue.

In contrast, model estimates have more uncorrelated natural variability: totally uncorrelated for coupled models; and largely uncorrelated for SST-forced models (the predictability of annual mean precipitation given perfect SSTs is low in eastern Africa). We approximate these correlations by taking the natural variability to be uncorrelated. The spread of model results can be compared by the spread expected by the natural variability by computing the $\chi^2/\mathrm{dof}$ statistic. If this is greater than one, there is a noticeable model spread, which is added in quadrature to the natural variability. This is denoted by the white boxes in Fig. 5. The bright red bar indicates the total uncertainty of the models, consisting of a weighted mean using the (uncorrelated) uncertainties due to natural variability plus an independent common model spread added to the uncertainty in the weighted mean.

Finally, observations and models are combined into a single result in two ways. Firstly, we neglect model uncertainties beyond the model spread and compute the weighted average of models and observations: this is indicated by the magenta bar. However, we know that models in general struggle to represent the climate of eastern Africa, so the model uncertainty is larger than the model spread. Therefore, secondly, we use the more conservative estimate of an unweighted average of observations and models, indicated by the white box around the magenta bar. This gives more weight to the observations.

### 3.1 List of assumptions/decisions

Following on from the chosen methodology, and assuming that, following model-evaluation, the data used is of sufficient quality, some further assumptions and decisions are made concerning the data/model setup.

1. We analyse Jan–Dec annual means. Based on the seasonal cycles of precipitation and temperature, for all regions except for region WE (which has a single rainy season) we could also have chosen to analyse Jul–Jun annual means instead. The influence of this choice on the trends is low (see also Sect. 3.2).

2. We assume for precipitation that the CenTrends dataset is better than other datasets over our region of study, as it includes many different sources of precipitation data (Funk et al., 2015). We therefore only use the CenTrends dataset for observations of precipitation.

3. We do not know of a realistic soil moisture dataset that covers a long-enough time period to calculate trends. Therefore we do not select simulations based on evaluation criteria other than selecting runs based on PET and precipitation evaluation in the input variables.

4. Trends are calculated or extrapolated using all data up to 2018 and between the pre-industrial era (1900) and the present (2018). Weather@home is an exception where trends are calculated between two stationary climates of the present and the pre-industrial era. Differences in trends can arise due to different time periods and lengths of datasets, which are generally shorter for observations and reanalyses than for model simulations.



5. In general we use the longest time series of data available. We make exceptions in the starting year if necessary, based on visual inspection of abrupt changes due to data limitations toward the beginning of the time series

    (a) We use Berkeley from 1900 and in region SS from 1920.

    (b) We use CRU starting from 1940 instead of 1901 in regions NK, CK and SS.

6. For consistency in the method, we fit the variability as a constant over time for all data. In both observed time series and simulations we see very little or no trend in variability up to 2018.

7. If for observational data a Gaussian fit is the best fit, we also fit model data to a Gaussian, even if a GPD is a better fit for that data. In doing this we avoid erroneous comparisons between the variable mean and variable extreme. We checked for model runs in which this disparity occurs, but found that in most cases the trend calculated from fitting model data to a GPD was not very different from the trend calculated from fitting model data to a Gaussian.

8. As models do not share a consistent set of soil moisture levels, we take the top level of each model, assuming that this is the most comparable level across models. We checked for LPJmL — the only selected ISIMIP hydrological model that has more than one level available for soil moisture — that the variability does not change by much when integrating over multiple levels instead of using level 1.

9. We focus on the historical time-frame. Therefore the trends in different RCP and socio-economic scenarios will be relatively similar to each other. The forcing data is the same for the years 1860–2005 and only differs for the most recent years, from 2006 onwards. In general, however, using different scenarios can be seen as an advantage, as a greater range of scenario uncertainty will be spanned.

    (a) We use RCP6.0 in ISIMIP as this choice resulted in the largest number of simulations, and RCP8.5 in EC-Earth as this was the only scenario available.

    (b) The socio-economic scenario selected in ISIMIP model runs is *historical*, for 1860–2005, and *2005soc* for 2006–present, except for H08 for which *historical* was not available for years 1860–2005 and we instead use *2005soc* for those years as well as years 2006–present. For the WFDEI experiments, *2005soc* was not available. Instead we use *varsoc* for the years 2006–2018, and *historical* before 2006.

10. Instead of PET, RefET(reference evapotranspiration) was available for the MERRA dataset. RefET can be converted to potential evapotranspiration by multiplying its value to a reference crop coefficient. We assume using RefET instead of PET does not influence the overall conclusions and we do not convert RefET into PET.

11. Within the ISIMIP project, variables required by the hydrological models, including temperature and precipitation, were bias corrected and the adjusted data was used to calculate PET and to drive the hydrological models to output soil moisture. In the synthesis, however, we present results for temperature and precipitation based on the unadjusted data, on principle that this better spans the range of model uncertainty in temperature and precipitation. The bias correction



applied in ISIMIP aims to conserve the original trend. In accordance, we find little change in trend for most time series, see Section 5.

## 3.2 Relating the results to recent droughts

People often initially experience droughts as reduced or failed rainy seasons. To portray the type of droughts we discuss and to relate the results to real events, we calculated return periods and risk ratios of recent droughts defined as low-precipitation events on the annual time scale, see Table 2. Note that the risk ratios are calculated from CenTrends alone and are not synthesized values based on a multi-model analysis. The synthesis of observations with models follows in the next section. We choose events based on the Emergency Events Database (EM-DAT) — an extensive global database of the occurrence and effects incurred from extreme weather events, and the time series calculated from CenTrends (up to December 2014 only, which excludes the recent droughts of 2015 and 2016/2017). For the northern three study regions we choose the year 2009, in which the first rainy season failed (in region WE, where there is only one peak in precipitation, the whole season had slightly lower precipitation amounts). For the southern three study regions we choose the year 2005, in which the second rainy season failed. Additionally, we also investigate the well-known 2010/2011 drought for the regions NK and SS. As this drought occurred over the latter part of 2010 and the first part of 2011 (the second part of 2011 was in fact very wet), we define the annual period of this specific 2010/2011 analysis to be Jul–Jun.

The results show, for instance for region WE, that in CenTrends the trend in precipitation between 1900 and 2018 is -0.09 mm/day/K (95% confidence interval (CI) -0.51 to 0.14 mm/day/K). With a change in GMST of 1.07 K and a mean precipitation in 1900 of 3.2 mm/day this is similar to a change of 3%. This means that if an event with the same precipitation amount as in the year 2009 had happened again in 2018 it would have been a one in 30 (95% CI 2 to 400) year event in 2018, whereas in 1900 it would have been a one in 80 (95% CI 30 to 1400) year event, corresponding to a probability ratio of 2.5 (95% CI 0.2 to 380). A return period that decreases in time indicates that such extreme droughts are becoming slightly more common, however, in this example we see large uncertainties consistent with no change. Note that the trend and probability ratio are not significantly different from zero at $p < 0.05$. The results for all regions are summarized in Table 2. We note that the trends calculated for the Jan–Dec events and for the Jul–Jun events in regions NK and SS respectively are not significantly different. This supports the decision to analyze Jan–Dec annual extremes only.

## 4 Synthesis results

In this section synthesis figures are presented for the region SS for each of the four variables. See the caption of Figure 5 for more information. The synthesis figures of all regions can be found in the Supplementary Information. Table 3 and Fig. 6 summarize all findings.

First we look for consistent behaviour in the trends from individual GCMs across the four variables. Some general conclusions looking at the different GCMs are as follows: (i) for GCM-driven model runs with stronger positive trends in temperature, there is a tendency that the positive trends in PET are also stronger, and vice versa; (ii) the uncertainty in precipitation trends is





**Table 2.** Trends, return periods and probability ratios of equivalent events in the year 2018 and 1900 for three recent drought events registered in the EM-dat database (2005, 2009 and 2010/2011), based on annual average precipitation (mm/day) from the CenTrends dataset. 95 % confidence intervals are given between brackets. For each study region impacted by the events, the annual precipitation for the event year ($Prcp$, used to define the event magnitude) and the 1900–2014 climatological precipitation average ($\overline{ClimPrcp}$) is given. The asterisk (*) denotes that Jul–Jun is taken instead of Jan–Dec to define a year.

| Region | Event year | $Prcp$ | $\overline{ClimPrcp}$ 1900-2014 | Trend [mm/dy/K] | Return period in 2018 | Return period in 1900 | Probability ratio |
|---|---|---|---|---|---|---|---|
| WE | 2009 | 2.94 | 3.38 | -0.09 (-0.51 to 0.14) | 30 (2 to 400) | 80 (30 to 1400) | 2.5 (0.2 to 380) |
| EE | 2009 | 1.49 | 1.84 | -0.03 (-0.35 to 0.07) | 40 (3 to 340) | 50 (25 to 560) | 1.4 (0.4 to 70) |
| NS | 2009 | 0.42 | 0.63 | 0.07 (-0.08 to 0.12) | 80 (4 to 300) | 10 (5 to 46) | 0.13 (0.03 to 6.7) |
| NK | 2005 | 0.77 | 1.10 | -0.07 (-0.26 to 0.12) | 5 (2 to 30) | 10 (5 to 22) | 1.9 (0.3 to 6.5) |
| CK | 2005 | 1.75 | 2.39 | 0.04 (-0.55 to 0.43) | 29 (3 to 200) | 22 (12 to 63) | 0.77 (0.11 to 14) |
| SS | 2005 | 0.74 | 1.09 | 0.03 (-0.12 to 0.22) | 29 (4 to 470) | 17 (6 to 47) | 0.61 (0.02 to 7.80) |
| WK | 2010/ 2011* | 0.51 | 1.10 | 0.16 (-0.30 to 0.27) | 650 (10 to 20000) | 130 (53 to 2200) | 0.21 (0.03 to 64) |
| SS | 2010/ 2011* | 0.53 | 1.09 | 0.02 (-0.31 to 0.21) | 300 (12 to 40000) | 230 (90 to 8100) | 0.77 (0.03 to 80) |

high compared to the trend magnitudes. This is one of the reasons why a clear relation with soil moisture trends is not evident; (iii) no clear relation between local temperature trends and soil moisture trends is evident.

Looking at the different hydrological models we conclude that the trend in PCR-GLOBWB PET, which uses the Hamon PET scheme that depends only on temperature, is generally higher than the trend in in EC-Earth PET, which uses the more-complex Penman-Monteith PET scheme that additionally depends on humidity, wind and radiation. Using this more complex scheme can influence the trend in soil moisture, especially in wetter regions.

The analyses of the individual model runs, stratifying by GCM or hydrological model, do not lead to a clear conclusion on the relation between the trends in precipitation, temperature, PET and soil moisture. We therefore turn to the analysis of the synthesized values, see Table 3 and Fig. 6 for a summary of the outcome, and Fig. 5 and Figs. S1 to S6 in the Supplementary Information for synthesis diagrams.





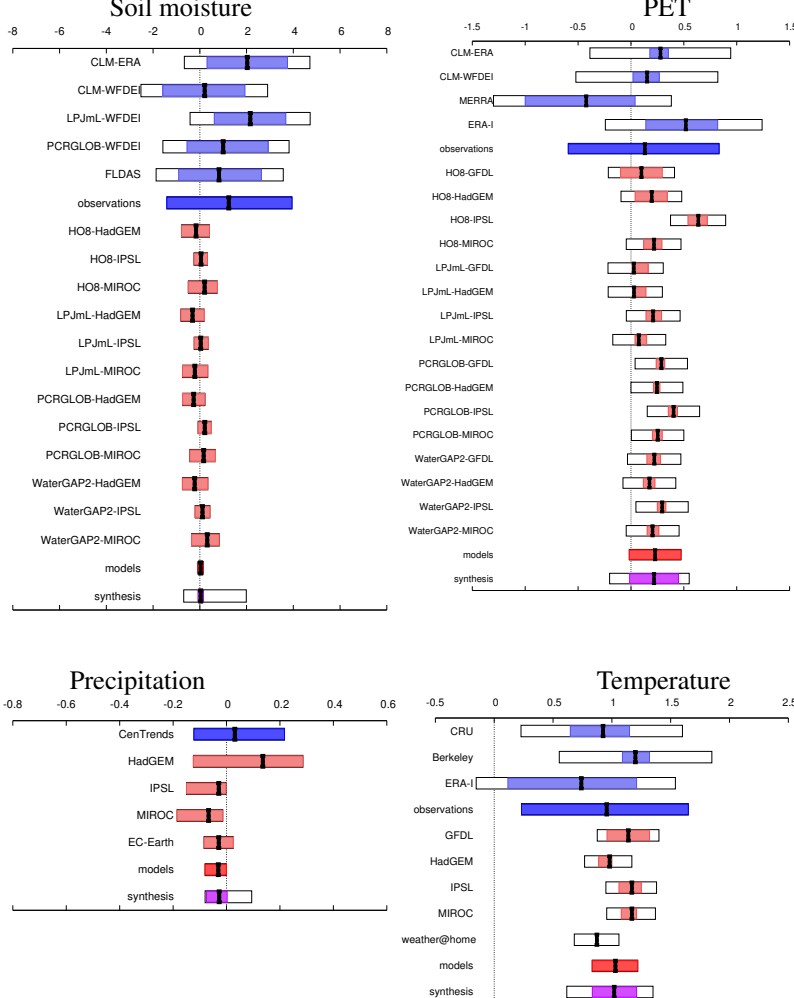

**Figure 5.** Illustrative examples of the synthesized values of trends per degree GMST rise for soil moisture (top left), PET (top right), precipitation (bottom left) and temperature (bottom right) for region SS. Black bars are the average trends, colored boxes denote the 95% CI. Blue represents observations and reanalyses, red represents models and magenta the weighted synthesis. Coloured bars denote natural variability, white boxes also take representativity / model errors into account if applicable (see Sect. 3). In the synthesis, the magenta bar denotes the weighted average of observations and models and the white box denotes the unweighted average. Soil moisture trends are based on standardized data, the other trends are absolute trends.

For soil moisture we find no significant synthesized trends: there is practically no change in region EE, and no trend to a small positive non-significant trend in regions WE, NS, NK, CK and SS.

For precipitation, regions WE and NK show a positive but non-significant trend, in region NS there is a small positive trend, regions EE and CK show no trend and region SS a negative non-significant trend.





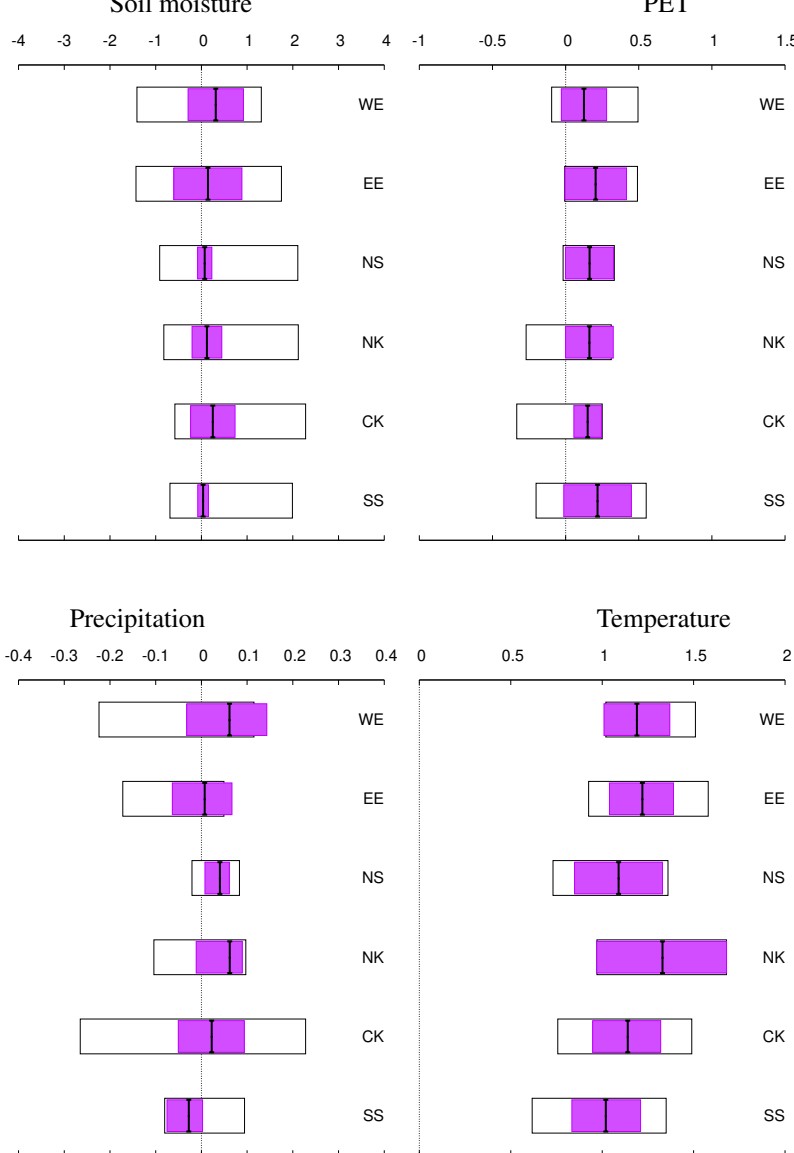

**Figure 6.** Summary of the synthesized values for soil moisture, PET, precipitation and temperature in the six regions. The magenta bars denote the weighted averages of observations and models and the white boxes denote the unweighted averages.

As expected from global climate change, the local annually averaged temperatures all have a significant positive trend, with best estimates between 1.0°and 1.3°per degree of GMST increase. Related to this, trends in PET are also positive in four of the six regions but lower than for temperature and generally with larger confidence intervals. The exceptions are regions NK and CK, where, although the weighted averages show positive trends, the models show opposite tendency to observations and

5      are thus incompatible with them, rendering the result uncertain.





**Table 3.** Summary of synthesis results for each region and study variable. Note that '0' means no *significant* change and a '+' sign indicates a positive trend, where in soil moisture this means a change towards a *wetter* soil.

| Region | Soil moisture | Precipitation | Temperature | PET |
|---|---|---|---|---|
| WE | 0/+ | 0/+ | + | + |
| EE | 0 | 0 | + | + |
| NS | 0/+ | + | + | + |
| NK | 0/+ | 0/+ | + | 0/+ |
| CK | 0/+ | 0 | + | 0/+ |
| SS | 0/+ | 0/- | + | + |

We can identify the following relationships between different variables: (i) Precipitation trends have a (small) influence on soil moisture trends in regions WE, NS and NK; (ii) in regions WE, EE, NS, NK and CK, temperature and PET have no discernible influence on soil moisture trends; (iii) in region SS, the non-significant negative trend in precipitation does not lead to lower soil moisture, and neither do temperature increase or the trend in PET.

## 5 Discussion

In this section, we discuss ways in which our chosen approach to studying drought in eastern Africa may have influenced the results obtained.

We focus on drought on annual time scales, as long-term drought presents a greater risk for food security, with the annual period defined from January to December. This definition is a natural choice for each of our study regions, where the single or dual seasonal cycle peaks in precipitation (rainy seasons) and temperature do not extend beyond December into the next year. The Jan–Dec definition has the consequence that multi-season droughts out of phase with this period, such as the well-documented 2010/2011 event in eastern Africa that affected only the second rainy season in 2010 and first rainy season of 2011, do not appear extreme in the observational time series used here, whilst they would appear extreme in a Jul–Jun series. This does however not affect the annual trends in the region which are similar independent of defining the year from Jan–Dec or from Jul–Jun. Hence our choice of annual definition does not significantly influence the results.

On the annual time scale, we do not see strong explanatory relationships between the *trends* in the four studied variables. To gain insight in the relationships between the variables, we additionally looked at correlations on a sub-annual time scale. Simple correlations between monthly precipitation, temperature, PET, and soil moisture (not shown) support the conclusions of Manning et al. (2018) on the influence of precipitation and PET on soil moisture at dry sites. They found that at water-limited sites the influence of precipitation on soil moisture is much larger than the influence of temperature, via PET, on soil moisture. In our study, we find the same for the driest regions and the driest months in the wetter regions, and for the more temperature-based PET schemes.





Looking at seasonal cycles — monthly means averaged over recent decades — a comparison between seasonal cycles of the different variables shows that the seasonal cycle of soil moisture is similar to that of precipitation in all six study regions. In contrast, the inverse seasonal cycle of temperature is not similar to that of soil moisture. Whether the PET seasonal cycle reflects elements of the soil moisture cycle or not depends on the PET scheme used: temperature- or radiation-based schemes

show a seasonal cycle that is similar to that of temperature, whereas more advanced schemes reflect a mixture between the seasonal cycles of precipitation and temperature.

We thus conclude that the influence of precipitation on soil moisture is higher than that of temperature or PET. This is supported by the synthesized results that show negligible or no trends in soil moisture and precipitation, whereas the trends in temperature and PET are strongly positive.

If temperature has, via PET, an influence on trends in soil moisture, we expect to see that the positive trend in temperature is coupled to a drying trend in soil moisture. As we average over the annual scale, we may miss parts of the season when this effect is strongest. Therefore we selected a region and period outside the rainy season, in which the seasonal peak in temperature corresponds to a dip in soil moisture (region CK, months Feb–Mar), to inspect sub-annual trends (not shown). Even then, we find that there is no negative trend in soil moisture accompanying the positive temperature trends.

The approach taken in this paper towards uncertainty has been to

- Perform a multi-model and multi-observation analysis that summarises what we know at the present moment, using readily available data and methods.

- Apply simple evaluation techniques to readily available data, treating datasets that satisfy evaluation criteria equally and rejecting the others.

- Communicate uncertainties from synthesis. A simple 'yes' or 'no' is not appropriate in this analysis where there is no clear significant positive or negative trend. Rather, the uncertainties and their origin are given.

We find that the PET scheme used has a large influence on the PET seasonal cycle and trend (Prudhomme et al., 2014). In any case, in the long term, a trend in PET only has meaning for crop growth if there is water available for evaporation. Much of eastern Africa is in a water-limited evaporation regime. In the case that irrigation would be locally applied, more

water would become available for evaporation, shifting the situation away from a water-limited regime and towards an energy limited regime. A trend in PET seen in our analyses (especially if the analysis using different schemes produces a robust PET trend) could then signify a trend in real evaporation and would therefore be accompanied by an increase in irrigation water demand. Note that irrigation is not accounted for by the models or reanalysis datasets used here.

Rowell et al. (2015) discussed the possibility that climate model trends are influenced by inability of the models to represent

key physical processes reliably, and flagged this issue as a topic for further study. In attribution studies on drought, especially for this region, it is therefore high priority to extend model evaluation techniques to assess models' representation of key physical processes. The approach taken in this paper has been to apply simple evaluation techniques to readily available data, in order to advance our current knowledge. Precipitation seasons in this region are governed by large-scale processes,



such as the shifting of the ITCZ, and ENSO dynamics. The ability of a model to capture the seasonal cycle in precipitation thus provides some assurance that large-scale physical processes are reasonably well described by the model. The frequency of extremes in precipitation is influenced by variability with respect to the general magnitude, therefore we also check that variability in analysed time series is similar to observations. We see these tests as a minimum requirement for model validation.

However, to improve the performance of models and to understand the discrepancies between models and observations, a much more thorough investigation into the models' representation of physical processes and feedbacks is required, such as demonstrated by James et al. (2018) and encouraged by the IMPALA (Improving Model Processes for African Climate) project (https://futureclimateafrica.org/project/impala/).

While improving the data with respect to some characteristics, an additional uncertainty arises from the bias correction of the

GCM data prior to use in the hydrological model. The bias correction in ISIMIP was set up to preserve the long-term trend, but it also decreases the daily variability by truncating extreme high values (e.g., in precipitation) (Hempel et al., 2013). The most important element for our analysis is that it also increases the daily variability by removing excessive drizzle, which is often present in GCM precipitation data. Prudhomme et al. (2014) noted that such a statistical bias correction can influence the signal of runoff changes but that the effect generally remains smaller than the uncertainty from GCMs and global impact models. By

far the largest difference we found in our analysis between trends in original and bias-corrected data was for temperature for IPSL in region NK: we found 1.9 K/K (95% CI 1.8 to 2.1 K/K) for the original trend and 1.4 K/K (95% CI 1.3 to 1.5 K/K) for the trend in bias-corrected data. All other differences were smaller and non-significant.

It is also important to note the significant scientific uncertainty relating to the effects of increased atmospheric $CO_2$ concentrations on the relationship between PET and plant growth and so on drought.

There is some evidence that warm spells are increasing in length, particularly in Ethiopia and northern Somalia/Somaliland region (Gebrechorkos et al., 2019), as is the number of consecutive dry days in some parts of eastern Africa, which may have an impact on drought length and increase the onset rapidity and intensity of drought (Trenberth et al., 2014). However the overall impact on crops and food security during long-duration droughts on annual timescales is probably insensitive to this.

It is possible that increasing temperatures have a negative impact on food security during droughts in ways that are be-

yond the scope of this study, e.g., decreased immunity of livestock, or increased water demand for cooling and water supply (Gebrechorkos et al., 2019, and references therein). In addition, in regions suffering from recent meteorological drought, non-meteorological factors such as increasing population and land-use changes also play a role in worsening the declining vegetation conditions, even after precipitation returns to normal (Pricope et al., 2013).

## 6 Conclusions

Previous attribution studies for the eastern Africa region have examined drought from a meteorological perspective (precipitation deficit), and have found no clear trends above the noise of natural variability. In this study, we examined drought from an agricultural perspective (soil moisture) as well as the meteorological perspective, and additionally investigated whether



increasing global and local temperatures and trends in evaporative demand can be seen to contribute to trends towards drier soils.

Using a combination of models and observational datasets we studied trends in four drought related annual variables — soil moisture, precipitation, PET, and temperature — for six regions in eastern Africa. In this section, we draw conclusions for each variable in turn. Out of the four studied variables, food security is most closely related to soil moisture anomalies. In standardized soil moisture data, we find no discernible trends. The uncertainties in trends from model runs are large, and there are no long observational runs available. This makes the use of an ensemble of models imperative. Due to the large uncertainties in both soil moisture observations and simulations, soil moisture cannot be relied upon on its own as a drought indicator and it is therefore important to examine other drought indicators as well. Besides, soil moisture also has a physical lower limit: once the soil is dry it will remain dry. In water limited regions an analysis of precipitation is thus a helpful addition.

Precipitation was found to have a stronger influence than temperature or PET on soil moisture variability, especially in the drier study regions (the significant positive trend in temperature is not reflected by a decrease in soil moisture). However, the error margins on precipitation trend estimations are large and no clear trend is evident.

As expected from the increase in global temperatures, we find significant positive trends in local temperatures in all six regions. The synthesized trend is between 1.0 and 1.3 times the trend in GMST, which corresponds to a local temperature rise of 1.1 to 1.4 degrees from pre-industrial times to 2018. However, the influence of this on annual soil moisture trends appears limited.

PET has a more direct link via evaporation to soil moisture than temperature. The trends in PET are predominantly positive, although in the regions NK and CK the value is uncertain. This generally agrees with the positive trends in temperature. Similar to the results for temperature, we do not find strong relations between PET and soil moisture trends. Nevertheless, the results can still be of interest, especially in irrigated regions. Note, however, that there are generally large differences between results from different model runs due to the different PET schemes and input datasets used. An analysis of PET should therefore be carried out using a model ensemble that (i) covers various PET schemes, especially if the additional drivers required for the more advanced PET schemes are not monitored well, and (ii) uses multiple input datasets to span the uncertainty from driving GCMs. Conclusions on resulting trends should be drawn with caution, giving more weight to the confidence interval around the best estimate rather than the best estimate alone.

Considering food security, a drought becomes more problematic if it endures for more than one season. So although the trends in seasonally averaged variables may be larger than those in annually averaged variables, the conclusions drawn for annual droughts are of higher relevance for food security.

Whilst it may be preferable to use soil moisture as a drought indicator, observations and simulations of precipitation are more reliable. Precipitation has a large influence on agricultural droughts and is therefore appropriate to use in attribution studies in eastern Africa, supplementing the analysis of soil moisture. The outcome of previous studies that have focussed on precipitation deficits only (e.g., Philip et al., 2018a; Uhe et al., 2018) are thus still relevant and compare well with our results here, that no consistent significant trends on droughts are found.



Finally, communication of the uncertainties in the analyses of soil moisture, precipitation, temperature, and PET (and any drought indicators) to policy makers, the media and other stakeholders is crucial. Without insight into the uncertainties in synthesized trends in the different drought indicators, conclusions become meaningless and results can easily be misinterpreted.

*Data availability.* Almost all time series used in the analysis are available for download under https://climexp.knmi.nl/EastAfrica_timeseries.
cgi (last access: 29 April 2019).

*Author contributions.* Sarah Kew and Sjoukje Philip designed and coordinated the study, analysed all data and led the writing of the manuscript. Mathias Hauser contributed the CLM datasets including PET calculations and substantially contributed to writing. Mike Hobbins produced the RefET dataset and substantially contributed to writing. Niko Wanders and Karin van der Wiel collaborated to create the EC-Earth - PCR-GLOBWB data, including PET calculations and bias correction. Niko Wanders additionally advised on the use and validation
of PET and soil moisture data for the analysis of drought. Geert Jan van Oldenborgh contributed analysis tools, monitored progress and contributed to writing. Ted I.E. Veldkamp provided guidance on the use of ISIMIP data and contributed to discussions. Joyce Kimutai and Chris Funk provided local information. Friederike E.L. Otto conceived the idea for the study, monitored development, provided weather@home results and contributed to writing.

*Competing interests.* We declare that there are no competing interests.

*Acknowledgements.* We would like to acknowledge funding for this work from the Children's Investment Fund Foundation (CIFF) grant #1805-02753. For their roles in producing, coordinating, and making available the ISI-MIP model output, we acknowledge the modelling groups and the ISI-MIP coordination team. We would also like to thank the volunteers running the weather@home models as well as the technical team in OeRC for their support. MERRA-2 data were developed by the Global Modeling and Assimilation Office (GMAO) at NASA Goddard Space Flight Center under funding from the NASA Modeling, Analysis, and Prediction (MAP) program; the data are disseminated
through the Goddard Earth Science Data and Information Services Center (GES DISC), preliminary data have been made available thanks to GMAO.



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
