# Peer review of "Impact of precipitation and increasing temperatures on drought trends in Eastern Africa"

_Earth System Dynamics, 2019_

## Referee Comment (RC1) · Anonymous Referee #1 · 20 Jun 2019

This paper attempts to demonstrate trends in different hydroclimatic variables and how they may relate to recent droughts in eastern Africa. While it is laudable that many (mostly model-based) time series have been used to address uncertainties, the documentation of these data is somewhat confusing, the presentation of results lack clarity and the interpretation/discussion of findings is rather vague. In general, the material is presented in a way that makes it hard to follow the implications of the chosen synthesis method, the differences among models and regions, and the overall conclusions; also, rather long-term trends than droughts are being analysed.

Main comments:

Already the title is somewhat misleading, as you do not really analyse droughts but (modelled) annual soil moisture and climatic trends. I understand the argument that

soil moisture and PET may be proxies for agricultural drought, but the connection of this analysis to drought and even food security is too vague. Moreover, the analysis of long-term annual trends probably tells little about the (shorter-term) droughts. The attempt to interpret the recent drought years as part of the overall trends is too limited. I'd suggest to rephrase the setup in terms of that you analyse hydroclimatic trends over the study region rather than suggesting the analysis is on droughts. Moreover, the study may better fit a specialised hydrological or climatological journal.

Introduction: lenghty, with some passages that do not straightforwardly lead to the study's objectives or promise too much. Specifically, I think, the statements on food production (p. 3 lines 3-15) are not needed; it could be much more straightforwardly said that you analyse four variables without attempting to construct such an argument (which you instead could shortly point to in the Discussion/Conclusions). The paragraphs on p. 4 also belong rather to the Discussion. The study's leading questions should be much more concrete, and focused on the East African region.

Study region: It is not clear how the three criteria were applied: homogeneous precipitation (is it really homogeneous by the way, at what time scale?), livelihood zones, expert judgment? And is it really so that the final results are only aggregated for these 6 zones, and only based on annual data? This should be said clearly early on, as it limits the scope of the analysis (while arguably increasing robustness).

Datasets: It would be very helpful if there was a summary of the methodological approach in the very beginning of the Methods or as part of the Introduction. Figure 2 and the following text is not easy to follow; a well-structured and annotated table showing all data, acronyms, time periods and references would be way better.

Methods: Not clear to me why "global" temperature is used and what the purpose of this analysis is. Is it done for all time series, and why not just use the original data? What "validation tests" were done, and if they were more or less qualitative you may still have applied a quasi-objective criterion of whether the seasonal cycle "resembles" the

observational data (which actually). What is a return period for a specific year, 2018 (e.g. Fig. 3)? Why does w@h require no fitting?

Page 13 point 4: So different time periods are mixed in your synthesis product? Doesn't that produce biases or at least merit discussion?

Section 3.2 is weak; I do not see a convincing approach to drought analysis here, and why is but one illustrative example explained which also only says that there is a marginally significant trend over the whole time period? This seems to be also something that should go to the Results.

Results: I am not sure if this is the best selection of figures to portray results, and whether the set can be extended (note, the methods part has more figures than the main text's results part). Are also maps possible? Why focus text on the SS region only? In any case, the results section is way too short, and the reader gets lost on what figures, tables, data you refer to in the Results' text. A clearer presentation of key findings is needed, plus a more academic style (terms like "looking at" etc. should be more precise analytically). The order of presentation also need improvement, maybe variable after variable.

Discussion: rather a list of shortcomings (which does not build trust in the analysis) than a discussion of the main findings and their relevance. Surely every analysis has caveats, but in this paper the robust patterns need to be highlighted and then discussed in terms of their plausibility and potential further studies to be done as follow-up.

Conclusions: too long and not really conclusions but an extension of the Discussion.

Detailed/technical comments:

Abstract: Study period needs to be mentioned. Line 12, "Nevertheless...", this info is not needed here. Line 14/15, this is self-evident and no novel conclusion of this study I'd say.

Introduction: Line 30, GCM is the abbrevation for General Circulation Models. Page 4

lines 20-27: can be deleted

Datasets: what is the original spatial resolution of the different data, and how were they aggregated? W@home data: using the counterfactual climate dataset seems to make no sense here? Page 8 line 27: why not shown, what sort of analysis is this? Line 33: what is refET? Page 9 line 3-13: belongs to Discussion, as not studied hereand probably not relevant for the historical time period. Line 17: what is the relevance of the RCPs here, as you do not analyse future periods.

Discussion: page 19 line 17: where is this subannual analysis presented, and why not part of the Results (same for the analysis of PET differences, page 20 line 22)? Page 2o line 21: uncertainties and origin are given: not so clear, and this is also in contrast to what is presented in Table 3.

---

## Author Comment (AC1) · 1 Jul 2019

We approach this study from an attribution science perspective. We acknowledge that the reviewer might not familiar with this approach. It is therefore obvious that our approach needs to be better explained for the general reader. The reply below concerns the most important issues raised here (1. title/topic of analysis, 2. awkward presentation) to stimulate discussion and aid the rest of the revision process. We will produce a full point-by-point response and additionally address all minor issues when the interactive discussion is over.

1. THE ANALYSIS IS ON LONG-TERM TRENDS RATHER THAN DROUGHT

It is known that eastern African temperatures are increasing in step with global warm-

ing. The objective of the paper is to determine if increasing temperatures are exacerbating long-term agricultural drought, i.e. are annual-time scale droughts increasing in frequency and intensity as the temperature increases? And/or are there human-induced changes in the annual (lack of) precipitation that have more explaining power than local temperature in increasing drought severity? To answer these questions, we indeed assess trends (with respect to GMST) in drought indices. There are various definitions of drought and we explain that we analyse changes (in frequency/intensity) to agricultural drought, which is normally measured by the state of the soil, soil moisture. The title is therefore, in our opinion, not incorrect. We will, however, make explicit from the start that the paper is about long-term drought.

2. MATERIAL IS PRESENTED IN A WAY THAT MAKES IT HARD TO FOLLOW

We intend in our revisions to improve the presentation by:

* Stating more clearly that we will use an attribution approach to study agricultural drought in eastern Africa.

* Explaining clearly what we mean by an attribution approach and, in the methods, describing the steps involved which serve the final goal: arrive at one synthesized statement for each region. The steps contain: (i) the definition of the regions over which spatially and annually averaged time series are analysed (see Sect. 2.1), (ii) the detection of trends in observations with respect to GMST, accompanied by the calculation of a return period of a specific event threshold, (iii) model evaluation, (iv) calculation of trends in the models and (v) synthesis of the results. We will make clear that it is only in order to illustrate the method that we show some of the intermediate results in the methods section, whereas the synthesis results (the actual results of the study) are shown in Sect. 4. It will then be clear why, for example, we examine trends in GMST rather than in time, and do not map the results.

* Moving some topics from the introduction, as proposed by the reviewer e.g. removing discussion-like material and information on food security from the introduction to the

discussion/elsewhere.

* Clearly giving the motivation for choosing a particular structure to present results (why we first present synthesis diagrams region by region, and then comment on variable by variable).

* Clearly stating where examples are illustrative and not intended as results in response to the research question (sec 3.2). On a similar note - in section 4 (Synthesis of results) making clear that the text applies to all regions, not just region SS which is used as an illustrative example.

* Including a table as suggested to summarise the data sets used. We think preference for a table or a figure is personal (some people we consulted prefer to view the information in a figure), so propose to keep the figure in the main text and add a table to the supplementary, which can include additional information such as time periods available.

---

## Referee Comment (RC2) · Anonymous Referee #2 · 2 Dec 2019

**Comments for manuscript: esd-2019-20**

**Title**: Impact of precipitation and increasing temperatures on drought in eastern Africa
**Authors**: Sarah F. Kew, Sjoukje Y. Philip, Mathias Hauser, Mike Hobbins, Niko Wanders, Geert Jan van Oldenborgh, Karin van der Wiel, Ted I.E. Veldkamp, Joyce Kimutai, Chris Funk, and Friederike E.L. Otto

**Summary**

In this paper, the authors obtained the sensitivities of soil moisture, precipitation, potential evapotranspiration (PET) and local temperature to global mean surface temperature (GMST) from numerous datasets using statistical tools, and tried to explain the trend in soil moisture as a combination of trends in precipitation and potential evapotranspiration in eastern Africa.

I believe that the authors did a lot of work to quantify the synthesized values of sensitivities, which may be helpful for the drought analysis of eastern Africa. However, as far as I am concerned, the writing of this manuscript need significant improvement, for example, the logical chain of the paper is poor; some expressions are not appropriate (i.e., temperatures?) and can be confusing to understand. Thus, as scientific research, it does need substantial improvements to presents a sufficiently significant advance to meet the ESD standards.

**Major points:**

1. The logical chain of the paper seems to be incorrect. The target of this paper is to investigate the impact of precipitation and temperature on drought, which, however, was not quantified in the paper. In fact, the authors only showed the sensitivity of soil moisture, precipitation, PET and local temperature to GMST without any details of the physical mechanism.

2. The method to quantify the sensitivity of different variables to GMST is unclear in the paper. As shown in line 27, page 9, "The method is extensively explained in van Oldenborgh et al. (2019) and Philip et al. (2019)", however, van Oldenborgh et al. (2019) is in review (line 29, page 28) and Philip et al. (2019) is in preparation (line 22, page 27). Therefore, I believe it's better to illustrate some necessary mechanism of the method in the paper.

3. As shown in Figure 3b, the 95% confidence interval for fitted location parameter of precipitation to GMST is quite large. I wonder how precise the sensitivity of precipitation to GMST in the paper is since even the paper itself referred to the fact that "the effect of a changing climate on precipitation is generally much less straightforward" in line 17, page 2.

4. The authors claim to use as many datasets as readily available, provided that the data are sufficiently complete over a long-enough time period. Moreover, there are different hydro/impact models being applied to simulate PET and SM. Two questions are raised here, first, since the accuracy of different datasets may vary spatially, is it reasonable to use as many datasets as readily available, particularly, without applying any additional bias correction (as suggested in Page 6 line 2); second, a very long paragraph is organized here to describe different projects and models, however, differences among these models are not highlighted and the reasons why these projects and models were selected are not clear. Section 2.2 needs serious revisions.

5. The result that 'Precipitation has a stronger influence on soil moisture variability than temperature or PET in the drier or water-limited region' seems to be one of the major conclusions in this study. In fact, there are studies revealing the fact that precipitation is more influential on soil moisture over dry regions and temperature is more influential on soil moisture in wet regions. The authors may need to highlight the novelty of this study in different ways.

**Some specific points:**

1. Page 1 line 5, we studied trends in six regions or four drought-related variables? I suppose they refer trends in four drought-related variables, however, the statement is not appropriate.

2. Page 4 lines 6-15, this paragraph doesn't seem to be closely related to the topic of this manuscript?

3. Page 4 line 24, a discussion and conclusions are… this is suggested to be changed to discussions and conclusions are…

4. Page 4 line 27, in this section we show… this sentence can be moved to Line 22 before in Section 3 to keep consistency and avoid a one-sentence paragraph.

5. Table 1 and Figure 1 have basically the same information, no need to keep both. Suggest keeping only Figure 1. The authors mentioned that six regions are selected based on livelihood, precipitation zones and local expert judgment, suggest clarifying these criteria clearly in Table 1.

6. In Table 3, the description of '-' (a negative trend) is missing. A small comment, I think table 3 may not be necessary here since similar information has been conveyed in Fig. 6.

7. Lines 6-7 in page 19, not clear

8. Page 20 line 22, we find that … (Prudhomme et al., 2014). It is not clear whether the conclusion comes from the author or from other's work.

9. Page 20 line 31, it is therefore (of) high priority? And line 32…..has been to apply simple, these sentences seem problematic to me.

10. The inconsistency between Figure and Fig. in the manuscript (e.g. Page15 lines 27-28).

11. For soil moisture and precipitation, both low extremes are targeted. Why the distribution functions are different?

12. Are there any proofs suggesting that the CenTrends precipitation dataset is better than others?

13. Content in Section 3.1 is hard to follow due to the poor logic. Suggest reorganizing.

---

## Author Comment (AC2) · 26 Mar 2020

**Final author comments**

**Comments from Referees**

**Reviewer Comment 1**

**This paper attempts to demonstrate trends in different hydroclimatic variables and how they may relate to recent droughts in eastern Africa. While it is laudable that many (mostly model-based) time series have been used to address uncertainties, the documentation of these data is somewhat confusing, the presentation of results lack clarity and the interpretation/discussion of findings is rather vague. In general, the material is presented in a way that makes it hard to follow the implications of the chosen synthesis method, the differences among models and regions, and the overall conclusions; also, rather long-term trends than droughts are being analysed.**

*Response:* Thank you for your insightful review. We have made substantial changes to the structure and have hopefully improved the readability and clarity of the manuscript.

**Main comments:**

**Already the title is somewhat misleading, as you do not really analyse droughts but(modelled) annual soil moisture and climatic trends. I understand the argument that soil moisture and PET may be proxies for agricultural drought, but the connection of this analysis to drought and even food security is too vague. Moreover, the analysis of long-term annual trends probably tells little about the (shorter-term) droughts. The attempt to interpret the recent drought years as part of the overall trends is too limited. I'd suggest to rephrase the setup in terms of that you analyse hydroclimatic trends over the study region rather than suggesting the analysis is on droughts. Moreover, the study may better fit a specialised hydrological or climatological journal.**

*Response:* Thank you for your comment. We now see that the title does not reflect the paper content well enough. In general 'droughts' are defined as 'below normal water', and when examining soil moisture this can be referred to as 'agricultural droughts'. From this perspective, we think the use of the word drought in the title and study is

appropriate, but we propose to also add the word 'trends' to the title as indeed the study concerns trends in drought rather than drought itself or specific episodes. We will also emphasize that while shorter-term drought can be severe, we choose to analyse trends in longer-term (annual) drought as the impacts are far reaching. Concerning the connection to food security, we will edit the paragraph at the top of page 3 - see the response to the following reviewer point. Section 3.2 entitled 'Relating the results to recent droughts' (now renamed as "Illustrative examples") is actually intended as a presentation of illustrative examples of the method and not as an interpretation of the results. We want to show that known recent droughts that local readers will be familiar with do stand out (have multiple year return periods) following the annual averaging and that the starting month of the annual averaging has no influence on the resulting trend detected. We propose to change the subsection title and will also explain better the purpose of the subsection.

*Changes:*
Manuscript Title: Impact of precipitation and increasing temperatures on drought **trends** in eastern Africa
In the abstract, p.1 L3: In the current study we focus on **trends in long-term** agricultural drought
Section 3.2:
      title changed to: Illustrative examples
      See also responses below to questions on Methods section.

**Introduction: lenghty, with some passages that do not straightforwardly lead to the study's objectives or promise too much. Specifically, I think, the statements on food production (p. 3 lines 3-15) are not needed; it could be much more straight forwardly said that you analyse four variables without attempting to construct such an argument(which you instead could shortly point to in the Discussion/Conclusions).**

*Response:* Thank you, we see your point and we will reduce the length of the introduction and make it more focussed on the study's objectives.
*Changes:* We will remove lines 5-11 from p.3. The paragraph will then be re-written. The purpose of the paragraph will be to briefly motivate the choice of study variables and express the wish to align the study variables as closely as possible with one of the major impacts of drought - reduced food security.

**The paragraphs on p. 4 also belong rather to the Discussion. The study's leading questions should be much more concrete, and focused on the East African region.**

*Response:* We agree. The section from p.3 L29 - p4. L15 is indeed lacking focus and contains elements that would be better in the discussion. Reviewer 2 is of the opinion that p.4 L6-14 (probably s/he means L6-15) are not closely related to the topic of the manuscript and we agree to remove that paragraph.

*Changes:* The paragraph will be used to highlight the use of PET and soil moisture in previous drought attribution/trend studies, with less or no detail on the outcomes of individual studies. p.4 L6-15 will be removed. The rest of this chunk will be moved to the discussion and also condensed.
The leading questions have been edited, as specified below.

**Study region: It is not clear how the three criteria were applied: homogeneous precipitation (is it really homogeneous by the way, at what time scale?), livelihood zones, expert judgment? And is it really so that the final results are only aggregated for these6 zones, and only based on annual data? This should be said clearly early on, as it limits the scope of the analysis (while arguably increasing robustness).**

*Response:* We have to strike a balance between the size of the regions and their homogeneity and did so under guidance of local experts. The annual mean precipitation as well as the seasonal cycle in precipitation is used to assess homogeneity in precipitation. For example, the WE box coincides with the wettest part of Ethiopia and is clearly distinct from the EE box in annual precipitation. Its southern boundary is fixed however at 7N rather than further south because south of this latitude the form of the seasonal cycle changes from a single to double peak in precipitation. The broad livelihood zones displayed in Fig. 1b were then consulted to check that the land use in the chosen regions was predominantly the same kind(s) and that the boundaries make sense. Box EE is the least homogenic, but we decided to keep this the same as in a separate already published study on Ethiopia (Philip et al., 2015), also because it was discussed with local experts. Experts from the National Meteorological Agency of Ethiopia (in Philip et al., 2015) and FEWS NET also reviewed and discussed our chosen regions in terms of homogeneity.
The final results are indeed based on annual time scales, with conclusions drawn for the six study regions individually. Averaging over large regions indeed makes results more robust. In our study region, averaging over smaller regions would result in too much

uncertainty in the results and climate models would be less able to capture the small areas. Longer droughts, spanning more than one growing season, have an impact on food security, and therefore we average over the year. This again is a compromise - the more growing seasons affected by drought, the larger the impact, but averaging over multiple years reduces the length of the data series and increases confidence intervals. Therefore we indeed aggregate over a time interval of one year and over each chosen region. We will make this clearer in the text.

*Changes:* We selected six regions based on precipitation zones *in which the seasonal cycle is homogeneous (Fig. 1a),* livelihood zones *(see Fig. 1b),* and *discussions with local experts from Kenya Meteorological Department, and the National Meteorological Agency (NMA) of Ethiopia and* the Famine Early Warning Systems Network (*FEWS NET)*.
*Added:* Data is annually averaged and spatially averaged over the study regions.

**Datasets: It would be very helpful if there was a summary of the methodological approach in the very beginning of the Methods or as part of the Introduction. Figure 2 and the following text is not easy to follow; a well-structured and annotated table showing all data, acronyms, time periods and references would be way better.**

*Response:* we agree it would increase readability if we mention the methodological approach further towards the beginning of the paper, in the introduction. Furthermore we moved some text following Figure 2 to the supplement as it is not relevant for understanding the results, and we turned the list below Figure 2 into a table. We however received positive comments on the figure itself and prefer to keep both the figure and table of data including references. The figure has the advantage that the connections between the datasets, i.e. which driving model/data sets feed which hydrological/impact models, as well as the number of runs, are easily visualised. We agree however that the description of the data below the table decreases readability and therefore does not belong to the main paper. We moved this to the supplement.

*Changes:*
- We added a paragraph before the current last paragraph of the introduction that reads:
  We have a stepwise approach to answer the attribution question:
1. Definition of the event and explanation of the study regions
2. Description of observational data and detection of trends in observations
3. Model evaluation including description of the models
4. Attribution of trends in models
5. Synthesis of the results

- In the methods section we added tables for the observational and model data sets instead of the itemized list, in which we will specify the short and full name of each data set, the time period and reference:

| Observational dataset | Full Name | Time period | Reference |
|---|---|---|---|
| Gridded data set / reanalysis | | | |
| CenTrends (prcp) | ... | ... | ... |
| ... | | | |
| Observation-driven hydro/impact model | | | |
| LPJmL-WFDEI (soil moisture) | ... | ... | ... |
| ... | | | |

| Model data set | Full name | Time period | Reference |
|---|---|---|---|
| GCM/RCM | | | |
| GFDL (temp, prcp) | ... | ... | ... |
| ... | | | |
| Hydro/impact model | | | |
| H08 (soil moisture, PET) | ... | ... | ... |
| ... | | | |

- To the first paragraph of the methods section we added "Furthermore, the two subsections describe (i) the assumptions and decisions that are made concerning the data/model setup and (ii) an example of how the method is applied to real data."

**Methods: Not clear to me why "global" temperature is used and what the purpose of this analysis is. Is it done for all time series, and why not just use the original data? What "validation tests" were done, and if they were more or less qualitative you may still have applied a quasi-objective criterion of whether the seasonal cycle "resembles" the observational data (which actually). What is a return period for a specific year, 2018(e.g. Fig. 3)? Why does w@h require no fitting?**

*Response:*
- we use GMST as a measure of anthropogenic climate change rather than just a trend over time. This is a common approach in attribution science. We added this to the text.
- The text reads "we check that the seasonal cycle resembles that of at least one of the observational datasets, in both the number and the timing of peaks." We thus require that the models broadly reflect the observed seasonal cycle. As there are sometimes also differences between observational data sets and, unless we have very good reasons, we do not rank one better than another, we do not use more objective sophisticated tests on the seasonal cycle than this. The fit parameters for the fit to GMST are assessed more objectively, as explained in the text.
- Concerning the comment on return period, in the method section, we explain that we evaluate the fitted distribution for the years 1900 and 2018, which means that for any threshold we can calculate a return period for the climate of 1900 and the climate of 2018. We added a sentence in the paper to explain this.
- The w@h does not require fitting as the large amount of data available for that model permits a direct estimation of the trend. This is already written on p 10 lines 12-14.

*Changed:* "We use global mean surface temperature (GMST) as a measure for anthropogenic climate change for calculating trends. We calculate trends for..."
Furthermore, after the sentence "In each case, the fitted distribution is evaluated twice: once for the year 1900 and once for the year 2018." we added "This allows us to calculate the return period of an event if it would have happened in the year 1900 or in the year 2018."

**Page 13 point 4: So different time periods are mixed in your synthesis product? Doesn't that produce biases or at least merit discussion?**

*Response:* Indeed the data sets do not all have the same time periods, but the data is first extrapolated onto the same time period (1900-2018), as mentioned here, before it is synthesized. However, there is no best way to tackle the problem of mixed products: models are framed differently and observational data has different lengths. Alternatives would be to restrict to the longest data set or use data with a common (shorter) time period, or not synthesize results at all. But this goes against our ethos. Our goal is to produce an overarching statement representing what we can conclude from a representative range of different available methods and data - i.e. methods which could

each have been used by others individually to report potentially conflicting messages in response to the same attribution (research) question. We want to use as much information as possible; only using the longest dataset or chosing just one framing in order to avoid this would not lead to more robust results, it would rather lead to an incomplete attribution result. We added this to item 4.

*Added:* We consider the use of all available observational and reanalysis data despite different model framings to lead to a more complete and robust attribution statement.

**Section 3.2 is weak; I do not see a convincing approach to drought analysis here, and why is but one illustrative example explained which also only says that there is a marginally significant trend over the whole time period? This seems to be also something that should go to the Results.**

*Response:* this section is added to illustrate the method and show an example. All other data sets are analysed following similar steps. In case this was not become clear in this section, we added a sentence on this. The trend results from these examples are already present in the results section, contained in the synthesis figures. The synthesis of observed and modelled trends is the main result and basis for the conclusions. Therefore we do not think the illustrative examples here are better moved to the results section.

*Changes: This section will be retitled "Illustrative examples"*
*Added: In this section we show an example to illustrate the method of detection of trends in precipitation data, as* droughts are often initially experienced as reduced or failed rainy seasons.

**Results: I am not sure if this is the best selection of figures to portray results, and whether the set can be extended (note, the methods part has more figures than the main text's results part). Are also maps possible? Why focus text on the SS region only? In any case, the results section is way too short, and the reader gets lost on what figures, tables, data you refer to in the Results' text. A clearer presentation of key findings is needed, plus a more academic style (terms like "looking at" etc. should be more precise analytically). The order of presentation also need improvement, maybe variable after variable.**

*Response:*
- We thought carefully about the selection of figures before submission and now have done so again, but we still think our selection of figures provides a good balance between simply showing the information needed to understand our

method along with the final results, and showing an overwhelming number of figures including intermediate results. We chose to explain the method (including figures) in the main text rather than referring to the supplement, which would in our opinion reduce readability. A description of the chosen figures is as follows:
- Figure 1: regions and arguments for selection of the regions
- Figure 2: illustration of the datasets
- Figures 3 and 4: illustration of the method using two different types of data (transient and stationary)
- Figure 5: illustration of the synthesis method including intermediate synthesis results
  - Supplement showing all other intermediate synthesis results
- Figure 6: summary of all synthesis results
- Except for Figure 1 we do not show maps, as we are analysing time series of area-averaged quantities rather than spatial fields.
- As outlined in the text, the results in the "synthesis results" section are for all six regions. The synthesis figure 6 also shows these results for all six regions. It is just Figure 5 that focuses on the SS region, to illustrate the synthesis method that, for all regions and all variables, leads to the final synthesis statements. As the final synthesis statements are much more important than the step in between, that is shown in Figure 5, similar figures with intermediate results for the other five regions are only shown in the supplement. We direct the reader to the supplement to see these intermediate results.
- We have now made it clearer in the text that *intermediate* synthesis figures for all six regions can be found in the supplement, but intermediate synthesis figures are presented for one region (SS) in the main text to illustrate the synthesis method. As we already discuss findings per variable, we also added a couple of sentences outlining the structure of this section.

*Changes:*
The first paragraph of the synthesis section now reads: "In this section, to illustrate the synthesis method, intermediate synthesis figures are presented for the region SS for each of the four variables. See the caption of Fig. 5 for more information. The intermediate synthesis figures for all six regions can be found in the Supplementary Information. Table 3 and Fig.~6 summarize final synthesized findings for all regions. Using both the intermediate and final synthesis results we first draw conclusions based on different GCMs and hydrological models and then present conclusions per variable. "

**Discussion: rather a list of shortcomings (which does not build trust in the analysis)than a discussion of the main findings and their relevance. Surely every**

**analysis has caveats, but in this paper the robust patterns need to be highlighted and then discussed in terms of their plausibility and potential further studies to be done as follow-up.**

*Response:* The intended purpose of the discussion here (and in many other papers) is to discuss the main concerns and thus to what extent the reported results are sensitive to the choices and assumptions we have made, and to put the results in context of related studies. These choices and assumptions limit the study in the sense that they define what has been studied. They are not intended to be portrayed as shortcomings but rather as choices necessary (as in any study) to make it achievable, useful and appropriate, given the resources available.

We recognise, however, that the discussion lacks structure and is fragmented. As pointed out in this review process, other sections contain information more suited to the discussion. These paragraphs have been merged into the discussion and the structure will be sharpened up, with topics dealt with in a more logical order and long paragraphs condensed. With the revised structure the discussion is no longer a list of shortcomings but a more general overview of the context and the influence of our choices and assumptions on the results.  Discussed topics include:
- The choice and definition of annual averaging scale: is the January-December definition appropriate? Would a different conclusion be reached using a sub-annual time scale?
- The potential influence of bias-correction on trends
- Our choice of model evaluation techniques in the light of recommendations from literature
- Our chosen approach towards communicating uncertainty of results
- The influence of the PET scheme on PET trends and the interpretation of PET trends in a water-limited regime, considering related studies
- The influence of (dynamic) vegetation schemes on drought trends, considering related studies and recommendations
- Factors beyond the scope of this study that may impact drought severity and food security

**Conclusions: too long and not really conclusions but an extension of the Discussion.**
*Response:* While restructuring the discussion section we moved some text from the conclusions to the discussion (i.e.. the discussion of food security and the use of

different PET schemes). In our opinion, the conclusions are now better structured and contain appropriate information.

**Detailed/technical comments:**

**Abstract: Study period needs to be mentioned.**
*Response: Thank you for noticing.*
*Changes:* To p.1 L3 we added "In the current study we focus on **trends in long-term** agricultural drought".

**Line 12, "Nevertheless...", this info is not needed here.**
*Changes:* "Using a combination of models and observational datasets, we studied trends, *spanning the period from 1900 (to represent the pre-industrial era) to 2018, …*"

**Line 14/15, this is self-evident and no novel conclusion of this study I'd say.**

*Response:* Whilst this conclusion might not be surprising in the light of other studies for different regions, it remains an important conclusion for Eastern Africa. This study was requested because the question of whether increasing temperatures are exacerbating drought keeps recurring.

**Introduction: Line 30, GCM is the abbrevation for General Circulation Models.**

*Response: Thank you for noticing.*
*Changes:* The correct expansion has been added.

**Page 4 lines 20-27: can be deleted**

*Response:* We are not sure whether the reviewer is really referring to p4 or not. Assuming he/she is: although it is not essential, for clarity we prefer to keep the paragraph (p4. L20-25) outlining the paper here, although L20-22 have been edited following Reviewer 2 specific point 4.

**Datasets: what is the original spatial resolution of the different data, and how were they aggregated?**

*Response:* We will refer the reader to references for the spatial resolution of the different data. In spatial aggregation, land grid points are weighted proportionally by the area represented.

**W@home data: using the counterfactual climate dataset seems to make no sense here?**

*Added:* Trends are calculated by dividing the difference in the variables between the present day climate and the counterfactual climate by the difference in GMST in the model in these two ensembles.

**Page 8 line 27: why not shown, what sort of analysis is this?**

*Response:* We analysed these datasets with different schemes to check the findings of Trambauer et al. (2014) also applies to our data. This is therefore not a novel idea nor a new finding, and besides we can not draw strong conclusions based on this that are relevant for the current analysis. Showing all details will distract the reader from the main findings. We therefore only mention that we checked this, but do not intend to include results.

**Line33: what is refET?**
*Response:* refET is daily reference evapotranspiration as mentioned in the data section.

**Page 9 line 3-13: belongs to Discussion, as not studied here and probably not relevant for the historical time period.**
*Response:* Thank you.
*Changes:* We indeed moved this paragraph to the discussion and shortened it.

**Line 17: what is the relevance of the RCPs here, as you do not analyse future periods.**

*Response:* It is not totally clear to us to which page this refers. However, differences in RCPs can account for uncertainty in the results, also for the near past. Between 2006 and 2018, there was a substantial increase in GMST and some spread in RCPs. Of course the difference would be larger if the analysis had extended to future periods.

**Discussion:**

**page 19 line 17: where is this subannual analysis presented, and why not part of the Results (same for the analysis of PET differences, page 20 line 22)?**

*Response:* We produced many more figures than those shown in this paper, e.g., the subannual analysis, the influence of different PET schemes on trends, the influence of different PET schemes compared to input datasets, the influence of using Jul-Jun instead of Jan-Dec etc. We think that presenting these extra analyses would add too much detail. We therefore only present the main findings and report the most important conclusions from additional analyses in only a few sentences. In doing so we keep the focus on the main findings.

**Page2o line 21: uncertainties and origin are given: not so clear, and this is also in contrast to what is presented in Table 3.**

*Changed:*
-   We added: Rather, the uncertainties (confidence intervals) and their origin (e.g. natural variability or model spread) are given.
-   we added "The table gives a concluding interpretation of the synthesized results shown in Fig. 6."
-   We added to the caption of Table 3 "The uncertainties associated with each result is depicted in Fig. 6."

**Reviewer Comment 2**

**In this paper, the authors obtained the sensitivities of soilmoisture, precipitation, potential evapotranspiration (PET) and local temperature to global mean surface temperature (GMST) from numerous datasets using statistical tools, and tried to explain the trend in soil moisture as a combination of trends in precipitation and potential evapotranspiration in eastern Africa. I believe that the authors did a lot of work to quantify the synthesized values of sensitivities, which may be helpful for the drought analysis of eastern Africa. However, as far as I am concerned, the writing of this manuscript need significant improvement, for example, the logical chain of the paper is poor; some expressions are not appropriate (i.e., temperatures?) and can be confusing to understand. Thus, as scientific research, it does need substantial improvements to presents a sufficiently significant advance to meet the ESD standards.**

*Response:* Thank you for your thorough review. We will check the appropriateness of expressions used. We hope that with the responses given and the changes proposed will alleviate the main concerns and that the resulting revised manuscript will satisfy ESD standards.

**Major points:**

1. **The logical chain of the paper seems to be incorrect. The target of this paper is to investigate the impact of precipitation and temperature on drought, which, however, was not quantified in the paper. In fact, the authors only showed the sensitivity of soil moisture, precipitation, PET and local temperature to GMST without any details of the physical mechanism.**

*Response:* Thank you for drawing our attention to this. We now see that the title does not reflect the paper content well enough, and that our approach to assessing the link between trends in precipitation, temperature and drought is poorly expressed. Indeed we do not intend to examine the *mechanism* by which precipitation affects drought, but rather (i) to investigate if there is a signal of change in agricultural drought indicators and (ii) to investigate which global-warming driven trends in precipitation or local temperature explain any emerging trend in agricultural drought. We propose to add the word 'trends' to the title as indeed the study concerns trends in drought rather than drought itself or specific episodes. We hope this clarifies the confusion about the target. We also added this to the abstract. We will clarify that we do not intend to examine mechanisms but primarily to detect (necessarily using both observations and models) whether there are GMST-driven trends in drought indicators, including temperature and precipitation, and if trends in precipitation and/or temperature are related to trends in agricultural drought. To make the procedure in the method more evident we added a paragraph in the introduction explaining the steps in the method.

*Changes:*
Manuscript Title: Impact of precipitation and increasing temperatures on drought *trends* in eastern Africa
In the abstract, p.1 L3: In the current study we focus on *trends in long-term* agricultural drought
Introduction: we changed the sentence on the second objective to (ii) to investigate which global-warming driven trends in precipitation or local temperature explain any emerging trend in agricultural drought.

Introduction: added before the last paragraph:
We have a stepwise approach to assess the link between trends in precipitation, temperature and drought:

1. Definition of the event and explanation of the study regions
2. Description of observational data and detection of trends in observations
3. Model evaluation including description of the models
4. Attribution of trends in models
5. Synthesis of the results

2. **The method to quantify the sensitivity of different variables to GMST is unclear in the paper. As shown in line 27, page 9, "The method is extensively explained in van Oldenborgh et al. (2019) and Philip et al. (2019)", however, van Oldenborgh et al. (2019) is in review (line 29, page 28) and Philip et al. (2019) is in preparation (line 22, page 27). Therefore, I believe it's better to illustrate some necessary mechanism of the method in the paper.**

*Response:* the revision of both papers is nearly finalized but neither has yet been published, so we decided to additionally refer to two other published papers in which the method is also described well. These are van Oldenborgh et al. (2018) and van der Wiel et al. (2017).

van der Wiel, K., Kapnick, S. B., van Oldenborgh, G. J., Whan, K., Philip, S. Y., Vecchi, G. A., Singh, R. K., Arrighi, J., and Cullen, H.: Rapid attribution of the August 2016 flood-inducing extreme precipitation in south Louisiana to climate change, Hydrol. Earth Syst. Sci., 21, 897–921, https://doi.org/10.5194/hess-21-897-2017, 2017.

3. **As shown in Figure 3b, the 95% confidence interval for fitted location parameter of precipitation to GMST is quite large. I wonder how precise the sensitivity of precipitation to GMST in the paper is since even the paper itself referred to the fact that "the effect of a changing climate on precipitation is generally much less straightforward" in line 17, page 2.**

*Response:* the 95% confidence intervals are calculated using a non-parametric bootstrapping procedure, i.e., we repeat the fit a large number of times (1000) with samples of (covariate,observation) pairs drawn from the original series with replacement. This is discussed in the papers we refer to, but we now also added this to the papers. The effect of climate change is not straightforward as natural variability is simply very high. No trend is therefore yet emerging over noise.

*Changes:* in the methods section we added: "Confidence intervals (CI) are estimated using a non-parametric bootstrapping procedure."

4. **The authors claim to use as many datasets as readily available, provided that the data are sufficiently complete over a long-enough time period. Moreover, there are different hydro/impact models being applied to simulate PET and SM. Two questions are raised here, first, since the accuracy of different datasets may vary spatially, is it reasonable to use as many datasets as readily available, particularly, without applying any additional bias correction (as suggested in Page 6 line 2); second, a very long paragraph is organized here to describe different projects and models, however, differences among these models are not highlighted and the reasons why these projects and models were selected are not clear. Section 2.2 needs serious revisions.**

*Response:*
1. Generally, our approach to attribution studies is to use and synthesize data that could have been produced by different teams separately, and to arrive at a conclusion based on a range of models and different (but compatible) framings of the research (attribution) question. However, we do reject models that are not fit for purpose in the validation step. Generally, we take the data as it comes and ideally as it would have been used in individual method studies, therefore including any corrections already applied to the data but not applying any more. Furthermore, in this case, we do not need a bias correction on the mean, as we are only looking at trends.
2. We use ISIMIP data because the ISIMIP project provides readily available model output of the variables under investigation. This is complemented by other readily available model runs with different (but compatible) framings. We will explain this in the text. The aim is however not to show differences between models, but rather to get a more complete answer on the attribution question. Different (types of) models could lead to different conclusions. With a multi-method and multi-model set-up study we make the attribution result more robust and thus gain confidence in the result.

   We do however agree that for this purpose the section on data is rather long. We therefore moved part of the model descriptions to the supplement.

Changes:
After "we use as many datasets as readily available, provided that the data are sufficiently complete over a long-enough time period to be used for trend calculations"

(p5 line 6-7) we add "and, for model data, provided that the model data pass the validation tests".

5. **The result that 'Precipitation has a stronger influence on soil moisture variability than temperature or PET in the drier or water-limited region' seems to be one of the major conclusions in this study. In fact, there are studies revealing the fact that precipitation is more influential on soil moisture over dry regions and temperature is more influential on soil moisture in wet regions. The authors may need to highlight the novelty of this study in different ways.**

*Response:* We will express that this is the first multi-model attribution study on several drought estimates in a highly vulnerable area, addressing a recurring question on whether increasing temperatures exacerbate drought.

**Some specific points:**

1. **Page 1 line 5, we studied trends in six regions or four drought-related variables? I suppose they refer trends in four drought-related variables, however, the statement is not appropriate.**

   *Response:* We are not sure what the reviewer misunderstands here, or why s/he thinks the statement is not appropriate. The text does not read 'in six regions **or** four … variables' but 'in six regions **in** four … variables'. In case it is the formulation which is confusing, we propose to change the text.

   *Changes:* Using a combination of models and observational datasets, for six regions in eastern Africa we studied trends in four drought-related annually averaged variables.

2. **Page 4 lines 6-15, this paragraph doesn't seem to be closely related to the topic of this manuscript?**

   *Response:* we agree; the detail of this paragraph is more distracting than helpful. *Changes:* This paragraph is deleted.

3. **Page 4 line 24, a discussion and conclusions are... this is suggested to be changed to discussions and conclusions are…**

*Response:* we agree 'a discussion' sounds strange. We will change 'a' to 'the', rather than 'discussion' to 'discussions' as then we still use the exact words in the section titles.
*Changes:* The text will now read '... the discussion and conclusions are presented in …'

4.

5. **Page 4 line 27, in this section we show... this sentence can be moved to Line 22 before in Section 3 to keep consistency and avoid a one-sentence paragraph.**

*Response:* It would reduce consistency to remove the sentence as we have such an introductory sentence at the beginning of each section (note, this doesn't apply to subsections). A one-sentence paragraph should not in itself be a problem, however, it would make sense to reduce the information on Section 2 in the paper outline (in lines 20-22) and transfer that information to the beginning of Section 2.
*Changes:* 'The outline of the remainder of the paper is as follows: In Section 2 the chosen study regions are presented followed by a description of the datasets used in the study.'

'In this section, we present the chosen study regions in eastern Africa and the datasets used to provide the four different variables (soil moisture, precipitation, temperature and PET) to be analysed. Brief descriptions of the (modelling) projects from which the datasets originate are provided'.

6. **Table 1 and Figure 1 have basically the same information, no need to keep both. Suggest keeping only Figure 1. The authors mentioned that six regions are selected based on livelihood, precipitation zones and local expert judgment, suggest clarifying these criteria clearly in Table 1.**

*Response:* The reviewer is correct that there is overlap of information, however, we think it is helpful to retain both means of presenting the information. A map shows very quickly the spatial relation between the study regions but it is easier to read off their coordinates in a collective table. We now notice it would be better

to use the same names of livelihood type as in the key of Fig.1b. Local expert opinion did modify our original study zone borders, for example, the Kenyan Meteorological Department suggested a westward extension of the original NK box and an increased separation between the original box NK and CK, according to their understanding of climatological and agricultural zones in Kenya. Also, acting on advice of Ethiopians in an earlier study, we shifted the northern boundary of box EE from 14degN to 13degN. We think these details, however, are too much for the manuscript.

*Changes:* In table 1, make the 5th column conform to nomenclature in Fig1b, land-use type → livelihood zone, add column to summarise climatological precipitation for each region.

7. **In Table 3, the description of '-' (a negative trend) is missing. A small comment, I think table 3 may not be necessary here since similar information has been conveyed in Fig. 6.**

*Response:* True, there is a negative sign in the table, and only the positive sign is explained. Whilst similar information is conveyed in table 3 and Fig. 6., we would argue for keeping both. The table summarises our interpretation of the numerical results in Fig. 6. The table is our conclusion on whether or not there are significant changes in the four variables in each region. Fig. 6 shows the numbers behind these conclusions and more importantly illustrates the uncertainties associated with each.

*Changes:* We will explain the above, i.e. the purpose of table 3 and Fig. 6 in the text, as well as adding a description of the '-' sign in the caption of Table 3.
To the caption of table 3 we will add that 'the uncertainties associated with each result are depicted in Fig.6'.

8. **Lines 6-7 in page 19, not clear**

*Response:* Concerns the words "In this section, we discuss ways in which our chosen approach to studying drought in eastern Africa may have influenced the results obtained." In our opinion, the purpose of a discussion is to interpret the results in the light of (i) how choices that have been made impact the outcome, and (ii) how the results relate to those previous studies on similar topics. It is not obvious what the reviewer finds unclear in the lines mentioned. Either s/he does not think the sentence describes what we do in the discussion, or perhaps s/he doesn't understand what is meant by "chosen approach".

*Changes:* Incase the latter is true, we will change "chosen approach" to "choices and assumptions".

9. **Page 20 line 22, we find that ... (Prudhomme et al., 2014). It is not clear whether the conclusion comes from the author or from other's work.**

   *Response:* We see that the reason for including the reference is not clear at all. In a global study, Prudhomme et al., found that GIMs contribute more than GCMs to the uncertainty in projected changes in drought and the uncertainty associated with GIMs has been attributed to differences in the number and type of processes represented in the GIMs (e.g., water balance, energy balance) and to differences in the details of their implementations. They do not specifically talk about PET schemes, however, so the reference will be removed.

   *Changes:* Prudhomme reference removed.

10. **Page 20 line 31, it is therefore (of) high priority? And line 32.....has been to apply simple, these sentences seem problematic to me.**

    *Response:* "therefore high" or "therefore of high" are both fine so we can add "of". We cannot see anything grammatically incorrect with line 32 "The approach taken in this paper has been to apply simple evaluation techniques to readily available data, in order to advance our current knowledge.", however we can change it to "In order to advance our current knowledge, in this paper we applied simple evaluation techniques to readily available data".

11. **The inconsistency between Figure and Fig. in the manuscript (e.g. Page15 lines 27-28).**

    *Response:* thank you for noticing.
    *Changes:* we will change instances of Figure to Fig., except where sentences begin with Figure(s).

**12.**

13. **For soil moisture and precipitation, both low extremes are targeted. Why the distribution functions are different?**

*Response:* We alluded to this in point 7 in the list of assumptions (section 3.1, page 14, line 7-10) but we could be more explicit in the main text as to why we use a specific distribution for a specific variable.

*Changes:* After inspection of whether a Gaussian or a General Pareto Distributions fits the observational or reanalysis data best, we use the following distributions:

14. **Are there any proofs suggesting that the CenTrends precipitation dataset is better than others?**

*Response:* This is our second assumption, but we will change that sentence slightly.

*Changes:* As was shown by Funk et al. (2015), the CenTrends precipitation dataset includes many different sources of precipitation data and more stations than most other datasets. We therefore assume for precipitation that the CenTrends dataset is superior to other datasets over our region of study. We therefore only use the CenTrends dataset for observations of precipitation.

**Content in Section 3.1 is hard to follow due to the poor logic. Suggest reorganizing.**

*Response:* Thank you for your suggestion to re-order this list into an order that makes more sense. We reorganised the assumptions, starting with all assumptions related to data issues, observational data and model data, continuing with assumptions that could impact the trend and finishing with the assumptions made on the fits. In the old numbering, the order of the list is: 2, 5, 3, 8, 11, 10, 9, 4, 1, 6, 7.

---

## Author Response (AR1)

**Major revision 1 comments**

**Comments from Referees**

**Reviewer Comment 1**

**This paper attempts to demonstrate trends in different hydroclimatic variables and how they may relate to recent droughts in eastern Africa. While it is laudable that many (mostly model-based) time series have been used to address uncertainties, the documentation of these data is somewhat confusing, the presentation of results lack clarity and the interpretation/discussion of findings is rather vague. In general, the material is presented in a way that makes it hard to follow the implications of the chosen synthesis method, the differences among models and regions, and the overall conclusions; also,rather long-term trends than droughts are being analysed.**

*Response:* Thank you for your insightful review. We have made substantial changes to the structure and have hopefully improved the readability and clarity of the manuscript.

**Main comments:**

**Already the title is somewhat misleading, as you do not really analyse droughts but(modelled) annual soil moisture and climatic trends. I understand the argument that soil moisture and PET may be proxies for agricultural drought, but the connection of this analysis to drought and even food security is too vague. Moreover, the analysis of long-term annual trends probably tells little about the (shorter-term) droughts. The attempt to interpret the recent drought years as part of the overall trends is too limited. I'd suggest to rephrase the setup in terms of that you analyse hydroclimatic trends over the study region rather than suggesting the analysis is on droughts. Moreover, the study may better fit a specialised hydrological or climatological journal.**

*Response:* Thank you for your comment. We now see that the title does not reflect the paper content well enough. In general 'droughts' are defined as 'below normal water', and when examining soil moisture this can be referred to as 'agricultural droughts'. From this perspective, we think the use of the word drought in the title and study is

appropriate, but we have added the word 'trends' to the title as indeed the study concerns trends in drought rather than drought itself or specific episodes. We will also emphasize that while shorter-term drought can be severe, we choose to analyse trends in longer-term (annual) drought as the impacts are far reaching. Concerning the connection to food security, we have edited the paragraph at the top of page 3 - see the changes made in response to the following reviewer point. Section 3.2 entitled 'Relating the results to recent droughts' (now renamed as "Illustrative examples") is actually intended as a presentation of illustrative examples of the method and not as an interpretation of the results. We want to show (i) that known recent droughts, particularly those that local readers will be familiar with, still stand out as extreme (have multiple year return periods) following the annual averaging and (ii) that the starting month of the annual averaging has no influence on the resulting trend detected. In addition to changing the subsection title we also now explain better the purpose of the subsection.

*Changes:*
- Manuscript Title: Impact of precipitation and increasing temperatures on drought **trends** in eastern Africa
- In the abstract, p.1 L3: In the current study we investigate **trends in long-term** agricultural drought
- P3 L2-3 changed to: "Whilst short term single-season drought episodes can be severe, we choose to analyse changes in drought on annual rather than sub-annual time scales because the worst crises in food security in this region have occurred with multiple season droughts (Funk et al. 2015)."
- Title of subsection 3.2 changed to: Illustrative examples
- The explanation "in subsection 3.3 we provide an example of how the method is applied to real data." is now inserted at the beginning of section 3,

See also responses below to questions on Methods section.

**Introduction: lenghty, with some passages that do not straightforwardly lead to the study's objectives or promise too much. Specifically, I think, the statements on food production (p. 3 lines 3-15) are not needed; it could be much more straight forwardly said that you analyse four variables without attempting to construct such an argument(which you instead could shortly point to in the Discussion/Conclusions).**

*Response:* Thank you, we see your point and we have reduced the length of the introduction and made it more focussed on the study's objectives.
*Changes:* We removed distracting lines from this paragraph on p.3. The remainder of paragraph has also been re-written. The purpose of the paragraph is now to briefly

motivate the choice of study variables and express the wish to align the study variables as closely as possible with one of the major impacts of drought - reduced food security.

**The paragraphs on p. 4 also belong rather to the Discussion. The study's leading questions should be much more concrete, and focused on the East African region.**

*Response:* We agree. The section from p.3 L29 - p4. L15 is indeed lacking focus and contains elements that would be better in the discussion. Reviewer 2 is of the opinion that p.4 L6-14 (probably s/he means L6-15) are not closely related to the topic of the manuscript and we have now removed that paragraph. The leading questions have been edited, and now read as below.

*Changes:*
- The paragraph is now used to highlight the use of PET and soil moisture in previous drought attribution/trend studies, with less detail on the outcomes of individual studies. p.4 L6-15 have been removed. Some of the preceding text has been moved to the discussion and also condensed.
- "... the objectives of this study are to (i) consider the attribution question ``do increasing global temperatures contribute to drier soils and thus exacerbate the risk of agricultural drought (low soil moisture) in eastern Africa?'' and (ii) to investigate if global-warming driven trends in precipitation or local temperature via PET explain any emerging trend in agricultural drought."

**Study region: It is not clear how the three criteria were applied: homogeneous precipitation (is it really homogeneous by the way, at what time scale?), livelihood zones, expert judgment? And is it really so that the final results are only aggregated for these6 zones, and only based on annual data? This should be said clearly early on, as it limits the scope of the analysis (while arguably increasing robustness).**

*Response:* We have to strike a balance between the size of the regions and their homogeneity and did so under guidance of local experts. The annual mean precipitation as well as the seasonal cycle in precipitation is used to assess homogeneity in precipitation. For example, the WE box coincides with the wettest part of Ethiopia and is clearly distinct from the EE box in annual precipitation. Its southern boundary is fixed however at 7N rather than further south because south of this latitude the form of the seasonal cycle changes from a single to double peak in precipitation. The broad livelihood zones displayed in Fig. 1b were then consulted to check that the land use in

the chosen regions was predominantly the same kind(s) and that the boundaries make sense. Box EE is the least homogenic, but we decided to keep this the same as in a separate already published study on Ethiopia (Philip et al., 2015), also because it was discussed with local experts from the National Meteorological Agency of Ethiopia. Experts from the Kenya Meteorological Department and FEWS NET also reviewed and discussed our chosen regions in terms of homogeneity.

The final results are indeed based on annual time scales, with conclusions drawn for the six study regions individually. Averaging over large regions indeed makes results more robust. In our study area, averaging over smaller regions would result in too much uncertainty in the results and climate models would be less able to capture the small regions.

Longer droughts, spanning more than one growing season, have an impact on food security, and therefore we average over the year. This again is a compromise - the more growing seasons affected by drought, the larger the impact, but averaging over multiple years reduces the length of the data series and increases confidence intervals. Therefore we indeed aggregate over a time interval of one year and over each chosen region. We have edited the text to make this clearer.

Changes: We selected six regions based on precipitation zones, in which the annual mean precipitation and seasonal cycle are homogeneous (Fig. 1a), livelihood zones (see Fig. 1b) and discussions with local experts from Kenya Meteorological Department, and the National Meteorological Agency (NMA) of Ethiopia and the Famine Early Warning Systems Network (FEWS NET).
Added: Data is annually and spatially averaged over the study regions.

**Datasets: It would be very helpful if there was a summary of the methodological approach in the very beginning of the Methods or as part of the Introduction. Figure 2 and the following text is not easy to follow; a well-structured and annotated table showing all data, acronyms, time periods and references would be way better.**

*Response:* we agree it would increase readability if we mention the methodological approach further towards the beginning of the paper, in the introduction. Furthermore we moved the text about data projects following Figure 2 to the supplement as it is not relevant for understanding the results, and we turned the list below Figure 2 into a 2-part table. We however received positive comments on the figure itself and prefer to keep both the figure and table of data including references. The figure has the advantage that the connections between the datasets, i.e. which driving model/data sets feed which hydrological/impact models, as well as the number of runs, are easily

visualised. We agree however that the description of the data below the table decreases readability and therefore does not belong to the main paper. We moved this to the supplement.

*Changes:*
- We added a paragraph before the current last paragraph of the introduction that reads: Our approach to attribution comprises the following steps: (1) Definition of the study variables and explanation of the study regions, (2) Description of observational data and detection of trends in observations (3) Model evaluation including description of the models, (4) Attribution of trends in models, (5) Synthesis of the results.

- In the methods section we added tables for the observational and model data sets instead of the itemized list, in which we will specify the short and full name of each data set, the time period, spatial resolution and references.

| Observational dataset | Full name | Time period used | Spatial resolution (°lat x °lon) | Reference(s) |
|---|---|---|---|---|
| **Observatational/reanalysis data set** | | | | |
| CenTrends (prcp) | Centennial Trends data set | 1900–2014 | 0.1x0.1 | Funk et al. (2015) |
| CRU TS4 (temp) | CRU TS4.01 | 1901–2019 | 0.5x0.5 | Harris et al. (2014) |
| Berkeley (temp) | Berkeley Earth | 1750–2019 | 1.0x1.0 | Rohde et al. (2013b, a) |
| ERA-I | ERA-Interim | 1979–2019 | 0.5x0.5 | Dee et al. (2011) |
| **Observation-driven hydro/impact model** | | | | |
| LPJmL-WFDEI (soil moisture) | Lund-Potsdam-Jena managed Land - WATCH-Forcing-Data-ERA-Interim | 1971–2010 | 0.5 x 0.5 | Bondeau et al. (2007); Rost et al. (2008); Schaphoff et al. (2013); Weedon et al. (2014) |
| PCRGLOB-WFDEI (soil moisture) | PCRaster GLOBal Water Balance model - WATCH-Forcing-Data-ERA-Interim | 1971–2010 | 0.5 x 0.5 | Sutanudjaja et al. (2018); Weedon et al. (2014) |
| CLM-ERA-I (soil moisture, PET) | Community Land Model version 4 - ERA-Interim | 1979–2016 | 0.5 x 0.5 | Oleson et al. (2010) |
| CLM-WFDEI (soil moisture, PET) | Community Land Model version 4 - WATCH-Forcing-Data-ERA-Interim | 1979–2013 | 0.5 x 0.5 | Lawrence et al. (2011); Weedon et al. (2014) |
| FLDAS (soil moisture) | Famine Early Warning Systems Network (FEWS NET) Land Data Assimilation System | 1981–2018 | 0.1 x 0.1 | McNally et al. (2017) |
| MERRA Ref-ET (PET) | Modern-Era Retrospective analysis for Research and Applications Reference Evapotranspiration | 1980–2018 | 0.125 x 0.125 | Hobbins et al. (2018) |

| Model dataset | Full name | Time period used | Spatial resolution (°lat x °lon) | Reference(s) |
|---|---|---|---|---|
| **GCM/RCM** | | | | |
| GFDL | GFDL-ESM2M, Geophysical Fluid Dynamics Laboratory - Earth System Model 2M | 1861–2018 | 2.02x2.5 | Dunne et al. (2012, 2013) |
| HadGEM | HadGEM2-ES, Hadley Centre Global Environmental Model version 2-ES | 1859–2018 | 1.25x1.88 | Collins et al. (2011); Jones et al. (2011) |
| IPSL | IPSL-CM5A-LR, Institut Pierre Simon Laplace - CM5A-LR | 1850–2018 | 1.89x3.75 | Dufresne et al. (2013) |
| MIROC | MIROC5, Model for Interdisciplinary Research on Climate - version 5 | 1850–2018 | 1.4x1.4 | Watanabe et al. (2010) |
| EC-Earth | EC-Earth 2.3 | 1850–2018 | 1.12x1.125 | Hazeleger et al. (2012) |
| w@h (temp, prcp, soil moisture) | Weather@home | 2005–2016 and counterfactual climate | 0.11x0.11 | Massey et al. (2015); Guillod et al. (2017) |
| **Hydro/impact models** | | | | |
| H08 (soil moisture, PET) | H08 | 1861–2018 | 0.5x0.5 | Hanasaki et al. (2008a, b) |
| LPJmL (soil moisture, PET) | Lund-Potsdam-Jena managed Land model | 1861–2018 | 0.5x0.5 | Bondeau et al. (2007); Rost et al. (2008); Schaphoff et al. (2013) |
| PCRGLOB (soil moisture, PET) | PCRGLOB-WB, PCRaster GLOBal Water Balance model | 1861–2018 | 0.5x0.5 | Sutanudjaja et al. (2018) |
| WaterGAP2 (soil moisture, PET) | Water Global Analysis and Progress Model version 2 | 1861–2018 | 0.5x0.5 | Müller Schmied et al. (2016) |

- To the first paragraph of the methods section we added "Furthermore, in subsection 3.2 we describe the assumptions and decisions that are made concerning the data/model setup and in subsection 3.3 we provide an example of how the method is applied to real data."

**Methods: Not clear to me why "global" temperature is used and what the purpose of this analysis is. Is it done for all time series, and why not just use the original data? What "validation tests" were done, and if they were more or less qualitative you may still have applied a quasi-objective criterion of whether the seasonal cycle "resembles" the observational data (which actually). What is a return period for a specific year, 2018(e.g. Fig. 3)? Why does w@h require no fitting?**

*Response:*
- We use GMST as a measure of anthropogenic climate change and express changes in local variables with respect to GMST rather than just calculating a trend over time. This is a common approach in attribution science. We added this to the text. It is used for all transient model runs and observational time series.

- Concerning validation tests, the manuscript text reads "we check that the seasonal cycle resembles that of at least one of the observational datasets, in both the number and the timing of peaks." We thus require that the models broadly reflect the observed seasonal cycle of the observational data set(s). As there are sometimes also differences between observational data sets and, unless we have very good reasons otherwise, we do not rank one better than another. We do not use more objective sophisticated tests on the seasonal cycle than this. The fit parameters for the fit to GMST are assessed more objectively, as explained in the text.
- Concerning the comment on return period, in the method section, we explain that we evaluate the fitted distribution for the years 1900 and 2018, which means that for any threshold we can calculate a return period for the climate of 1900 and the climate of 2018. We added a sentence in the paper to explain this.
- The w@h does not require fitting as the large amount of data available for that model permits a direct estimation of the trend. This is already written on p 10 lines 12-14.

*Changed:* "We use global mean surface temperature (GMST) as a measure for anthropogenic climate change for calculating trends. We calculate trends for..."
Furthermore, after the sentence "In each case, the fitted distribution is evaluated twice: once for the year 1900 and once for the year 2018." we added "This allows us to calculate the return period of an event as if it would have happened in the year 1900 or in the year 2018."

**Page 13 point 4: So different time periods are mixed in your synthesis product? Doesn't that produce biases or at least merit discussion?**

*Response:* Indeed the data sets do not all have the same time periods, but the data is first extrapolated onto the same time period (1900-2018), as mentioned here, before it is synthesized. However, there is no best way to tackle the problem of mixed products: models are framed differently and observational data has different lengths. Alternatives would be to restrict to the longest data set or use data with a common (shorter) time period, or not synthesize results at all. But this goes against our ethos. Our goal is to produce an overarching statement representing what we can conclude from a representative range of different available methods and data - i.e. methods which could each have been used by others individually to report potentially conflicting messages in response to the same attribution (research) question. We want to use as much information as possible; only using the longest dataset or chosing just one framing in

order to avoid mixing different time periods would not lead to more robust results, it would rather lead to an incomplete attribution result. We added this to item 4.

*Added:* However, we consider the use of all available observational and reanalysis data and different model framings to lead to a more complete and robust attribution statement.

**Section 3.2 is weak; I do not see a convincing approach to drought analysis here, and why is but one illustrative example explained which also only says that there is a marginally significant trend over the whole time period? This seems to be also something that should go to the Results.**

*Response:* Section 3.2 was included in the manuscript to illustrate the method and show an example. All other data sets are analysed following similar steps. In case this was not clear enough in this section, we added a sentence to explain this. The trend results from these examples are already present in the results section, contained in the synthesis figures. The synthesis of observed and modelled trends is the main result and basis for the conclusions. Therefore we do not think the illustrative examples here are better moved to the results section.

*Changes: Section 3.2 will be retitled "Illustrative examples"*
*Added: In this section we show an example to illustrate the method of detection of trends in precipitation data, as* droughts are often initially experienced as reduced or failed rainy seasons.

**Results: I am not sure if this is the best selection of figures to portray results, and whether the set can be extended (note, the methods part has more figures than the main text's results part). Are also maps possible? Why focus text on the SS region only? In any case, the results section is way too short, and the reader gets lost on what figures, tables, data you refer to in the Results' text. A clearer presentation of key findings is needed, plus a more academic style (terms like "looking at" etc. should be more precise analytically). The order of presentation also need improvement, maybe variable after variable.**

*Response:*
  - We thought carefully about the selection of figures before submission and now have done so again, but we still think our selection of figures provides a good balance between simply showing the information needed to understand our method along with the final results, and showing all intermediate results (an overwhelming number of figures). We chose to explain the method (including

figures) in the main text rather than referring to the supplement, which would in our opinion reduce readability. A description of the chosen figures is as follows:

- Figure 1: regions and arguments for selection of the regions
- Figure 2: illustration of the datasets
- Figures 3 and 4: illustration of the method using the two different types of data used (transient and stationary)
- Figure 5: illustration of the synthesis method including intermediate synthesis results
    - Supplement showing all other intermediate synthesis results
- Figure 6: summary of all synthesis results

- Except for Figure 1 we do not show maps, as we are analysing time series of area-averaged quantities rather than spatial fields.
- As outlined in the text, the results in the "synthesis results" section are for all six regions. The synthesis figure 6 also shows these results for all six regions. It is just Figure 5 that focuses on the SS region, to illustrate the synthesis method that, for all regions and all variables, leads to the final synthesis statements. As the final synthesis statements are much more important than the step in between, that is shown in Figure 5, similar figures with intermediate results for the other five regions are only shown in the supplement. We direct the reader to the supplement to see these intermediate results.
- We have now made it clearer in the text that *intermediate* synthesis figures for all six regions can be found in the supplement, but intermediate synthesis figures are presented for one region (SS) in the main text to illustrate the synthesis method. As we already discuss findings per variable, we also added a couple of sentences outlining the structure of this section.

*Changes:*
The first paragraph of the synthesis section now reads: "In this section, to illustrate the synthesis method, intermediate synthesis figures, which not only show the overall synthesis but also the results for individual models, are presented for the region SS for each of the four variables. See the caption of Fig. 5 for more information. The intermediate synthesis figures for all six regions can be found in the Supplementary Information. Table 3 and Fig.~6 summarize final synthesized findings for all regions. Using both the intermediate and final synthesis results we first draw conclusions based on different GCMs and hydrological models and then present conclusions per variable. "

**Discussion: rather a list of shortcomings (which does not build trust in the analysis)than a discussion of the main findings and their relevance. Surely every analysis has caveats, but in this paper the robust patterns need to be highlighted**

**and then discussed in terms of their plausibility and potential further studies to be done as follow-up.**

*Response:* The intended purpose of the discussion here (and in many other papers) is to discuss the main concerns and thus to what extent the reported results are sensitive to the choices and assumptions we have made, and to put the results in context of related studies. These choices and assumptions limit the study in the sense that they define what has been studied. They are not intended to be portrayed as shortcomings but rather as choices necessary (as in any study) to make it achievable, useful and appropriate, given the resources available.

We recognise, however, that the discussion lacks structure and is fragmented. As pointed out in this review process, other sections contain information more suited to the discussion. These paragraphs have been merged into the discussion and the structure will be sharpened up, with topics dealt with in a more logical order and long paragraphs condensed. With the revised structure the discussion is in our opinion no longer a list of shortcomings but a more general overview of the context and the influence of our choices and assumptions on the results.  Discussed topics include:
- The choice and definition of annual averaging scale: is the January-December definition appropriate? Would a different conclusion be reached using a sub-annual time scale?
- The potential influence of bias-correction on trends
- Our choice of model evaluation techniques in the light of recommendations from literature
- Our chosen approach towards communicating uncertainty of results
- The influence of the PET scheme on PET trends and the interpretation of PET trends in a water-limited regime, considering related studies
- The influence of (dynamic) vegetation schemes on drought trends, considering related studies
- Factors beyond the scope of this study that may impact drought severity and food security

**Conclusions: too long and not really conclusions but an extension of the Discussion.**
*Response:* While restructuring the discussion section we moved some text from the conclusions to the discussion (i.e., the discussion of food security and the use of different PET schemes) and reworded some paragraphs to make it clear that they concern concluding recommendations based on results. In our opinion, the conclusions now contain appropriate information.

**Detailed/technical comments:**

**Abstract: Study period needs to be mentioned.**
*Response: Thank you for noticing.*
*Changes:* To p.1 L3 we added "In the current study we focus on **trends in long-term** agricultural drought".
To p.1 L5 we added "Using a combination of models and observational datasets, we studied trends, *spanning the period from 1900 (to represent the pre-industrial era) to 2018, …*"

**Line 12, "Nevertheless...", this info is not needed here.**
*Response and changes:* This line has been left in, as it is an implication of our results. However we removed "as evaporation is water limited" from the sentence "However, the influence of these on soil moisture annual trends appears limited as evaporation is water limited", as this was not a direct result of our study.

**Line 14/15, this is self-evident and no novel conclusion of this study I'd say.**

*Response:* Whilst this conclusion might not be surprising in the light of similar studies for different regions, it remains an important conclusion for Eastern Africa. This study was requested (by CIFF, the Children's Investment Fund Foundation) because the question of whether increasing temperatures are exacerbating drought keeps recurring.

**Introduction: Line 30, GCM is the abbrevation for General Circulation Models.**

*Response: Thank you for noticing.*
*Changes:* The correct expansion has been added.

**Page 4 lines 20-27: can be deleted**

*Response:* We are not sure whether the reviewer is really referring to p4 or not. Assuming he/she is: although it is not essential, for clarity we prefer to keep the paragraph (p4. L20-25), which is the outline of the paper, although L20-22 have been edited following Reviewer 2 specific point 4.

**Datasets: what is the original spatial resolution of the different data, and how were they aggregated?**

*Response:* We have added the spatial resolution of each data set used to the data set table (Table 2 and 3 in section 2.2 of the revised manuscript). In spatial aggregation, land grid box values are aggregated by a simple average over the grid boxes. Over the coast the grid box values are included in the simple average if more than 50% of the grid box covers land, and are weighted by half if the center of the grid box lies on the coast.

*Changed*: the spatial resolution of each data set has been added to Table 2 and 3.

**W@home data: using the counterfactual climate dataset seems to make no sense here?**

Thank you for the comment. Indeed the result from w@h is not a pure trend from a transient simulation but rather the difference in the variables between the present day climate and the counterfactual climate ensemble scenarios divided by the difference in GMST in the model in these two ensembles. We approach this study from an attribution perspective in which both transient simulations and factual/counterfactual simulations are commonly used. In essence we perform part of an attribution analysis i.e. the detection of trends in both observations and climate models but do not specifically link the study to a particular event severity. In the paper we now make clear how we calculate the trend from factual and counterfactual runs.

*Added:* P8. L10, now moved to supplement: "Trends are calculated by dividing the difference in the variables between the present day climate and the counterfactual climate by the difference in GMST in the model in these two ensembles."

**Page 8 line 27: why not shown, what sort of analysis is this?**

*Response:* We analysed these datasets with different schemes to check that the findings of Trambauer et al. (2014) also apply to our data. This is therefore not a novel idea nor a new finding, and besides we cannot draw strong conclusions based on this that are relevant for the current analysis. Showing all details will distract the reader from the main findings. We therefore only mention that we checked this, but do not intend to include results. Note that this paragraph has since been moved to the discussion section.

**Line33: what is refET?**

*Response:* refET is daily reference evapotranspiration as mentioned in the data section and expanded on in the supplement.

**Page 9 line 3-13: belongs to Discussion, as not studied here and probably not relevant for the historical time period.**

*Response:* Thank you.

*Changes:* We indeed moved this paragraph to the discussion and shortened it. Dynamical vegetation models indeed show significant changes in the future, but could also be responsible for some uncertainty in the modelled response of drought to climate change over recent years.

**Line 17: what is the relevance of the RCPs here, as you do not analyse future periods.**

*Response:* It is not totally clear to us to which page this refers. However, differences in RCPs can account for uncertainty in the results, also for the near past. Between 2006 and 2018, there was a substantial increase in GMST and some spread in RCPs. Of course the difference would be larger if the analysis had extended to future periods.

**Discussion:**
**page 19 line 17: where is this subannual analysis presented, and why not part of the Results (same for the analysis of PET differences, page 20 line 22)?**

*Response:* We produced many more figures than those shown in this paper, e.g., the subannual analysis, the influence of different PET schemes on trends, the influence of different PET schemes compared to input datasets, the influence of using Jul-Jun instead of Jan-Dec etc. We think that presenting these extra analyses would add too much detail. We therefore only present the main findings and report the most important conclusions from additional analyses in only a few sentences. In doing so we keep the focus on the main findings.

**Page2o line 21: uncertainties and origin are given: not so clear, and this is also in contrast to what is presented in Table 3.**

*Changed:*
- We added to p20. L21: Rather, the uncertainties (confidence intervals) and their origin (e.g. natural variability or model spread) are given.
- we added to p16. L10: "The table gives a concluding interpretation of the synthesized results shown in Fig. 6."
- We added to the caption of Table 3 on p19 "The uncertainties associated with each result are depicted in Fig. 6."

**Reviewer Comment 2**

**In this paper, the authors obtained the sensitivities of soilmoisture, precipitation, potential evapotranspiration (PET) and local temperature to global mean surface temperature (GMST) from numerous datasets using statistical tools, and tried to explain the trend in soil moisture as a combination of trends in precipitation and potential evapotranspiration in eastern Africa. I believe that the authors did a lot of work to quantify the synthesized values of sensitivities, which may be helpful for the drought analysis of eastern Africa. However, as far as I am concerned, the writing of this manuscript need significant improvement, for example, the logical chain of the paper is poor; some expressions are not appropriate (i.e., temperatures?) and can be confusing to understand. Thus, as scientific research, it does need substantial improvements to presents a sufficiently significant advance to meet the ESD standards.**

*Response:* Thank you for your thorough review. We have checked the appropriateness of expressions used (our manuscript has been internally reviewed by an American native speaker) and have we made changes where we think necessary or where confusion may have resulted. Note that the expression "temperatures" is commonly used when referring to temperature in a general sense, e.g. "the effects of high temperatures during ..." is fine. We hope that with the responses given and the changes proposed will alleviate the main concerns and that the resulting revised manuscript will satisfy ESD standards.

**Major points:**

1. **The logical chain of the paper seems to be incorrect. The target of this paper is to investigate the impact of precipitation and temperature on drought, which, however, was not quantified in the paper. In fact, the authors only showed the sensitivity of soil moisture, precipitation, PET and local temperature to GMST without any details of the physical mechanism.**

*Response:* Thank you for drawing our attention to this. We now see that the title does not reflect the paper content well enough, and that our approach to assessing the link between the trends in precipitation, temperature and drought is poorly expressed.

Indeed we do not intend to examine the *mechanism* by which precipitation affects drought, but rather (i) to investigate if there is a signal of change in agricultural drought indicators and (ii) to investigate which global-warming driven trends in precipitation or local temperature explain any emerging trend in agricultural drought. We propose to add the word 'trends' to the title as indeed the study concerns trends in drought rather than drought itself or specific episodes. We hope this clarifies the confusion about the goal. We also added this to the abstract. We will clarify intend to detect (necessarily using both observations and models) whether there are GMST-driven trends in drought indicators, including temperature and precipitation, and if trends in precipitation and/or temperature are related to trends in agricultural drought. To make the procedure in the method more evident we added a paragraph in the introduction explaining the steps in the method.

*Changes:*
- Manuscript Title: Impact of precipitation and increasing temperatures on drought *trends* in eastern Africa
- In the abstract, p.1 L3: In the current study we focus on *trends in long-term* agricultural drought
- Introduction: we changed the sentence on the second objective to (ii) to investigate if global-warming driven trends in precipitation or local temperature via PET explain any emerging trend in agricultural drought.

Introduction: added before the last paragraph:
We have a stepwise approach to assess the link between trends in precipitation, temperature and drought:
1. Definition of the event and explanation of the study regions
2. Description of observational data and detection of trends in observations
3. Model evaluation including description of the models
4. Attribution of trends in models
5. Synthesis of the results

2. **The method to quantify the sensitivity of different variables to GMST is unclear in the paper. As shown in line 27, page 9, "The method is extensively explained in van Oldenborgh et al. (2019) and Philip et al. (2019)", however, van Oldenborgh et al. (2019) is in review (line 29, page 28) and Philip et al. (2019) is in preparation (line 22, page 27). Therefore, I believe it's better to illustrate some necessary mechanism of the method in the paper.**

*Response:* the revision of at least one of these papers is nearly finalized but neither has yet been published, so we decided to additionally refer to two other published papers in

which the method is also described well. These are van Oldenborgh et al. (2018) and van der Wiel et al. (2017).

*Changes:*
- "The method is extensively explained in Oldenborgh et al. (2019), Philip et al. (2019), Oldenborgh et al. (2018) and van der Wiel et al. (2017)."
- The second mention of references Oldenborgh et al. (2019) and Philip et al. (2019) on p9 L30 has been removed.

3. **As shown in Figure 3b, the 95% confidence interval for fitted location parameter of precipitation to GMST is quite large. I wonder how precise the sensitivity of precipitation to GMST in the paper is since even the paper itself referred to the fact that "the effect of a changing climate on precipitation is generally much less straightforward" in line 17, page 2.**

*Response:* the 95% confidence intervals are calculated using a non-parametric bootstrapping procedure, i.e., we repeat the fit a large number of times (1000) with samples of (covariate,observation) pairs drawn from the original series with replacement. This is discussed in the papers we refer to, but we now also added this to the manuscript. The effect of climate change is not straightforward to detect in such a complex region with complex climate dynamics and very large natural variability. The same holds for time as a covariate. No trend is therefore yet emerging over noise. We realised that we didn't add anything in the manuscript about the bootstrapping procedure, so we will add a sentence on that.

*Changes:* in the methods section we added: "Confidence intervals (CI) are estimated using a non-parametric bootstrapping procedure."

4. **The authors claim to use as many datasets as readily available, provided that the data are sufficiently complete over a long-enough time period. Moreover, there are different hydro/impact models being applied to simulate PET and SM. Two questions are raised here, first, since the accuracy of different datasets may vary spatially, is it reasonable to use as many datasets as readily available, particularly, without applying any additional bias correction (as suggested in Page 6 line 2); second, a very long paragraph is organized here to describe different projects and models, however, differences among these models are not highlighted and the reasons why these projects and models were selected are not clear. Section 2.2 needs serious revisions.**

*Response:*
1. Generally, our approach to attribution studies is to use and synthesize data that could have been produced by different teams separately, and to arrive at a conclusion based on a range of models and different (but compatible) framings of the research (attribution) question. However, we do reject models that are not fit for purpose in the validation step. Generally, we take the data as it comes and ideally as it would have been used in individual method studies, therefore including any corrections already applied to the data but not applying any more. Furthermore, in this case, we do not need a bias correction on the mean, as we are only looking at trends.

2. We use ISIMIP data because the ISIMIP project provides readily available model output of the variables under investigation. This is complemented by other readily available model runs with different (but compatible) framings. We have adjusted the text to explain this. The aim is however not to show differences between models, but rather to get a more complete answer on the attribution question. Different (types of) models could lead to different conclusions. With a multi-method and multi-model set-up study we make the attribution result more robust and thus gain confidence in the result.
We do however agree that for this purpose the section on data is rather long. We therefore moved part of the model descriptions to the supplement.

Changes:
- p5 L6-7: After "we use as many datasets as readily available, provided that (i) the data are sufficiently complete over a time period long enough to be used for trend calculations", we add "and (ii) the model data pass the validation tests".
- To p6 L5 we added: Many model simulations stem from the ISIMIP project, which provides output of the variables under investigation for four different impact models. These simulations are complemented by other readily available model runs with different (but compatible) framings.
- Information on model projects has been moved to the supplement.

5. **The result that 'Precipitation has a stronger influence on soil moisture variability than temperature or PET in the drier or water-limited region' seems to be one of the major conclusions in this study. In fact, there are studies revealing the fact that precipitation is more influential on soil moisture over dry regions and temperature is more influential on soil moisture in wet regions. The authors may need to highlight the novelty of this study in different ways.**

*Response:* We will express that this is the first multi-model attribution study on several drought estimates in a highly vulnerable area, addressing a recurring question on whether increasing temperatures exacerbate drought.

Changes: We added to the conclusions "In this first multi-model, multi-method attribution study using several drought estimates in eastern Africa, we address the recurring question on whether increasing global temperatures exacerbate drought."

**Some specific points:**

1. **Page 1 line 5, we studied trends in six regions or four drought-related variables? I suppose they refer trends in four drought-related variables, however, the statement is not appropriate.**

   *Response:* We are not sure what the reviewer misunderstands here, or why s/he thinks the statement is not appropriate. The text does not read 'in six regions **or** four … variables' but 'in six regions **in** four … variables'. In case it is the formulation which is confusing, we propose to change the text.

   *Changes:* Using a combination of models and observational datasets, for six regions in eastern Africa we studied trends in four drought-related annually averaged variables.

2. **Page 4 lines 6-15, this paragraph doesn't seem to be closely related to the topic of this manuscript?**

   *Response:* we agree; the detail of this paragraph is more distracting than helpful. *Changes:* This paragraph has been deleted.

3. **Page 4 line 24, a discussion and conclusions are... this is suggested to be changed to discussions and conclusions are…**

   *Response:* we agree 'a discussion' sounds strange. We will change 'a' to 'the', rather than 'discussion' to 'discussions' as then we still use the exact words in the section titles.

*Changes:* The text will now read '... the discussion and conclusions are presented in …'

4.

5. **Page 4 line 27, in this section we show... this sentence can be moved to Line 22 before in Section 3 to keep consistency and avoid a one-sentence paragraph.**

*Response:* It would reduce consistency to remove the sentence as we have such an introductory sentence at the beginning of each section (note, this doesn't apply to subsections). A one-sentence paragraph should not in itself be a problem, however, it would make sense to reduce the information on Section 2 in the paper outline (in lines 20-22) and transfer that information to the beginning of Section 2.
*Changes:*
- To the introduction: 'The outline of the remainder of the paper is as follows: In Section 2 the chosen study regions are presented followed by a description of the datasets used in the study.'

- At the beginning of section 2: 'In this section, we present the chosen study variables and study regions in eastern Africa and the datasets used to provide the variables to be analysed. Brief descriptions of the projects from which the datasets originate are provided in the supplement'.

6. **Table 1 and Figure 1 have basically the same information, no need to keep both. Suggest keeping only Figure 1. The authors mentioned that six regions are selected based on livelihood, precipitation zones and local expert judgment, suggest clarifying these criteria clearly in Table 1.**

*Response:* The reviewer is correct that there is overlap of information, however, we think it is helpful to retain both means of presenting the information. A map shows very quickly the spatial relation between the study regions but it is easier to read off their coordinates in a collective table. We now notice it would be better to use the same names of livelihood type as in the key of Fig.1b, so we have modified the last column of the table. Local expert opinion did modify our original study zone borders, for example, the Kenyan Meteorological Department suggested a westward extension of the original NK box and an increased separation between the original box NK and CK, according to their understanding of climatological and agricultural zones in Kenya. Also, acting on advice of

Ethiopians in an earlier study, we shifted the northern boundary of box EE from 14degN to 13degN. We think these details, however, are too much for the manuscript. Concerning the precipitation zones, there is non-overlapping information: the figure shows the spatial distribution of CenTrends annual average precipitation and in the table we now provide information on the seasonal cycle - single or dual peak month(s).

*Changes:* In table 1, we made the land-use column conform to nomenclature in Fig1b, and we rename land-use type as livelihood zone, and we added a column to summarise climatological seasonal precipitation cycle (titled seasonal precipitation peak(s)) for each region.

7. **In Table 3, the description of '-' (a negative trend) is missing. A small comment, I think table 3 may not be necessary here since similar information has been conveyed in Fig. 6.**

*Response:* True, there is a negative sign in the table, and only the positive sign is explained. Whilst similar information is conveyed in table 3 and Fig. 6., we would argue for keeping both. The table summarises our interpretation of the numerical results in Fig. 6. The table is our conclusion on whether or not there are significant changes in the four variables in each region. Fig. 6 shows the numbers behind these conclusions and more importantly illustrates the uncertainties associated with each.

*Changes:* We have added a description of the '-' sign in the caption of Table 3. To the caption of table 3 we will add that 'the uncertainties associated with each result are depicted in Fig.6'.

8. **Lines 6-7 in page 19, not clear**

*Response:* This remark concerns the words "In this section, we discuss ways in which our chosen approach to studying drought in eastern Africa may have influenced the results obtained." In our opinion, the purpose of a discussion is to interpret the results in the light of (i) how choices that have been made impact the outcome, and (ii) how the results relate to those previous studies on similar topics. It is not obvious what the reviewer finds unclear in the lines mentioned. Either s/he does not think the sentence describes what we do in the discussion, or perhaps s/he doesn't understand what is meant by "chosen approach".

*Changes:* Incase the latter is true, we will change "chosen approach" to "choices and assumptions".

9. **Page 20 line 22, we find that ... (Prudhomme et al., 2014). It is not clear whether the conclusion comes from the author or from other's work.**

   *Response:* We see that the reason for including the reference is not clear at all. In a global study, Prudhomme et al., found that global impact models (GIMs) contribute more than GCMs to the uncertainty in projected changes in drought and the uncertainty associated with GIMs has been attributed to differences in the number and type of processes represented in the GIMs (e.g., water balance, energy balance) and to differences in the details of their implementations. They do not specifically talk about PET schemes, however, so the reference will be removed.

   *Changes:* Prudhomme reference removed.

10. **Page 20 line 31, it is therefore (of) high priority? And line 32.....has been to apply simple, these sentences seem problematic to me.**

    *Response:* "therefore high" or "therefore of high" are both fine so we can add "of". We cannot see anything grammatically incorrect with line 32 "The approach taken in this paper has been to apply simple evaluation techniques to readily available data, in order to advance our current knowledge.", however we can change it as proposed below.

    *Changes*: "In order to advance our current knowledge, in this paper we applied simple evaluation techniques to readily available data".

11. **The inconsistency between Figure and Fig. in the manuscript (e.g. Page15 lines 27-28).**

    *Response:* thank you for noticing.

    *Changes:* we have changed instances of Figure to Fig., except where sentences begin with Figure(s).

12. **For soil moisture and precipitation, both low extremes are targeted. Why the distribution functions are different?**

*Response:* We alluded to this in point 7 in the list of assumptions (section 3.1, page 14, line 7-10) but we could be more explicit in the main text as to why we use a specific distribution for a specific variable.

*Changes:* After inspection of whether a Gaussian or a General Pareto Distribution fits the observational or reanalysis data best, we use the following distributions:

13. **Are there any proofs suggesting that the CenTrends precipitation dataset is better than others?**

*Response:* This is our second listed assumption, but we will change that sentence slightly.
*Changes:* As was shown by Funk et al. (2015), the CenTrends precipitation dataset includes many different sources of precipitation data and more stations than most other datasets. We therefore assume for precipitation that the CenTrends dataset is superior to other datasets over our region of study. We therefore only use the CenTrends dataset for observations of precipitation.

**Content in Section 3.1 is hard to follow due to the poor logic. Suggest reorganizing.**

*Response:* Thank you for your suggestion to re-order this list into an order that makes more sense. We reorganised the assumptions, starting with all assumptions related to data issues, observational data and model data, continuing with assumptions that could impact the trend and finishing with the assumptions made on the fits. In the old numbering, the order of the new list is: 2, 5, 3, 8, 11, 10, 9, 4, 1, 6, 7.

[revised manuscript text omitted]

 The outline of the remainder of the paper is as follows: In Section 2 the chosen study regions are presented followed by a description of the datasets used  in the study. In Section **??** we describe the  stepwise approach to attribution used in this paper, including  assumptions and decisions made and illustrative examples. In Section **??**, the results are synthesized per region. Finally,  the discussion and conclusions are presented in Sections **??** and **??**.

**2    Study variables, region and datasets**

[revised manuscript text omitted]

Within the second phase of the Inter-Sectoral Impact Model Intercomparison project (ISIMIP2 **?**) , global hydrological models are used for intercomparison of climate impacts. The model simulations use input from GCMs or observations, e.g., precipitation and PET. For more details on the ISIMIP project, please refer to **?** and **?** . We used modelled soil moisture (0.5× 0.5spatial resolution) for the period 1850/1861–2018 from four global hydrological models: H08 (**??**) , LPJmL (**???**) , PCR-GLOBWB (**???**) and WaterGAP2 (**?**) . The GCMs used are GFDL-ESM2M (GFDL, **??**) , HadGEM2-ES (HadGEM, **?**) , MIROC5 (MIROC, **?**) , IPSL-CM5A-LR (IPSL, **?**) . 
[revised manuscript text omitted]

| CRU TS4 (temp) | CRU TS4.01 | 1901–2019 | 0.5x0.5 | ? |
| Berkeley (temp) | Berkeley Earth | 1750–2019 | 1.0x1.0 | ?? |
| ERA-I | ERA-Interim | 1979–2019 | 0.5x0.5 | ? |
| **Observation-driven hydro/impact model** | | | | |
| LPJmL-WFDEI (soil moisture) | Lund-Potsdam-Jena managed Land - WATCH-Forcing-Data-ERA-Interim | 1971–2010 | 0.5 x 0.5 | ???? |
| PCRGLOB-WFDEI (soil moisture) | PCRaster GLOBal Water Balance model - WATCH-Forcing-Data-ERA-Interim | 1971–2010 | 0.5 x 0.5 | ?? |
| CLM-ERA-I (soil moisture, PET) | Community Land Model version 4 - ERA-Interim | 1979–2016 | 0.5 x 0.5 | ? |
| CLM-WFDEI (soil moisture, PET) | Community Land Model version 4 - WATCH-Forcing-Data-ERA-Interim | 1979–2013 | 0.5 x 0.5 | ?? |
| FLDAS (soil moisture) | Famine Early Warning Systems Network (FEWS NET) Land Data Assimilation System | 1981–2018 | 0.1 x 0.1 | ? |
| MERRA Ref-ET (PET) | Modern-Era Retrospective analysis for Research and Applications Reference Evapotranspiration | 1980–2018 | 0.125 x 0.125 | ? |

**Table 3.** Model data used in this study.

| Model dataset | Full name | Time period used | Spatial resolution (°lat x °lon) | Reference(s) |
|---|---|---|---|---|
| **GCM/RCM** | | | | |
| GFDL | GFDL-ESM2M, Geophysical Fluid Dynamics Laboratory - Earth System Model 2M | 1861–2018 | 2.02x2.5 | ?? |
| HadGEM | HadGEM2-ES, Hadley Centre Global Environmental Model version 2-ES | 1859–2018 | 1.25x1.88 | ?? |
| IPSL | IPSL-CM5A-LR, Institut Pierre Simon Laplace - CM5A-LR | 1850–2018 | 1.89x3.75 | ? |
| MIROC | MIROC5, Model for Interdisciplinary Research on Climate - version 5 | 1850–2018 | 1.4x1.4 | ? |
| EC-Earth | EC-Earth 2.3 | 1850–2018 | 1.12x1.125 | ? |
| w@h (temp, prcp, soil moisture) | Weather@home | 2005–2016 and counterfactual climate | 0.11x0.11 | ?? |
| **Hydro/impact models** | | | | |
| H08 (soil moisture, PET) | H08 | 1861–2018 | 0.5x0.5 | ?? |
| LPJmL (soil moisture, PET) | Lund-Potsdam-Jena managed Land model | 1861–2018 | 0.5x0.5 | ??? |
| PCRGLOB (soil moisture, PET) | PCRGLOB-WB, PCRaster GLOBal Water Balance model | 1861–2018 | 0.5x0.5 | ? |
| WaterGAP2 (soil moisture, PET) | Water Global Analysis and Progress Model version 2 | 1861–2018 | 0.5x0.5 | ? |

[revised manuscript text omitted]

290 to lower soil moisture  and neither do  the trends in temperature or PET.

**4 Discussion**

In this section, we discuss  the interpretation of our results in the light of how choices and assumptions made may have influenced the outcome and we compare previous studies on similar topics.

295    We study drought trends on annual as opposed to sub-annual time scales, as long-term drought presents a greater risk for food security . We define the annual period  to be from January to December. This definition is a natural choice for each of our study regions, where the single or dual seasonal cycle peaks in precipitation (rainy seasons) and temperature do not extend beyond December into the next year. The Jan–Dec definition has the consequence that multi-season droughts out of phase with this period  do not appear extreme in the observational time series used

300   here, whilst they would appear extreme in a Jul–Jun series. For example, in the well-documented 2010/2011 drought event in eastern Africa , only the second rainy season in 2010 and first rainy season of 2011  were exceptionally dry. This choice however does not affect the  resulting annual trends, which are similar  for both the Jan–Dec  and Jul–Jun does not significantly

305   annual definition.

On the annual time scale, we do not see strong explanatory relationships between the *trends* in the four studied variables. To gain insight in the relationships between the variables, we additionally looked at correlations on a sub-annual time scale. Simple correlations between monthly precipitation, temperature, PET  and soil moisture (not shown) support the conclusions of **?** on the influence of precipitation and PET on soil moisture at dry sites in Europe. They found that at water-limited sites the

[revised manuscript text omitted]

A study by **?** discussed the possibility that climate model  precipitation trends in East Africa are influenced by inability of the models to represent key physical processes reliably. In attribution studies on drought, especially for this region, it is therefore high priority to extend model evaluation techniques to assess models' representation of key physical processes. The approach taken in this paper has been to apply simple evaluation techniques  on the seasonal cycle and frequency distributions of readily available data  and that results from models passing validation tests represent the status of our current knowledge.  Rainy seasons in this region are governed by large-scale processes, such as the shifting of the ITCZ  and ENSO dynamics. The ability of a model to capture the seasonal cycle in precipitation and temperature thus provides some assurance that large-scale physical processes are reasonably well described by the model.  We see the tests we perform as a minimum requirement for model validation. However, to improve the performance of models and to understand the discrepancies between models and observations, a much more thorough investigation into the models'

representation of physical processes and feedbacks is required, such as demonstrated by **?** and encouraged by the IMPALA
(Improving Model Processes for African Climate) project (https://futureclimateafrica.org/project/impala/).

 It is still unknown how vegetation will respond to substantial increases of $CO_2$ concentration. Two counteracting effects — physiological (restriction of stomatal openings leading to decreased evapotranspiration) and structural (increased leaf area leading to more stomata and increased evapotranspiration) responses — are expected, but their net effect is unknown (e.g. **?**). So-called 'dynamic vegetation models' include these $CO_2$ effects and there are indications that these models show a weaker response of drought to climate change (**??**). In this study our selection of hydrological models is restricted by the variables we require, however, out of the four ISIMIP hydrological models that match our criteria, one (LPJmL) uses dynamic vegetation modeling. The soil moisture response to increasing GMST in LPJmL simulations is mid-range amongst the ISIMIP results. The PET response for LPJmL simulations is, however, somewhat on the low side of the ISIMIP results. It has not been verified if this behaviour is linked to dynamic vegetation modelling, but with confidence intervals generally overlapping with the synthesized model outcome, there is no exceptional difference.

[revised manuscript text omitted]

---

## Referee Report (RR1)

The authors have improved the manuscript and the logic is clearer. In general, the interpretation in the result section is still weak. There are still some issues that the authors may consider addressing or clarifying in the manuscript before I can recommend it for publication on ESD.

1. The novelty of this research, as highlighted by the authors, is the usage of multiple methods and models to investigate the impact of precipitation and temperatures on drought trends in EA. According to tables 2 and 3, observational/reanalysis and model data have different spatial resolutions, have they been resampled to the same resolution or used directly in the analysis? Also, the resolution of model data, e.g. GFDL (2.02°*2.5°), the extent of one region NK (2°N-4.5°N, and 34°E-41°E), it means probably only 6 grids are used to study the trends in variables in this region, whether the global-scale model simulation data is applicable to detect changes over small regions divided in table 1 / Figure 1. For trends derived from datasets with different resolutions, how were they eventually synthesized?

2. About datasets in section 2.2, the authors selected 35yrs or longer, multiple datasets spanning different lengths of years were used, e.g. observation Berkeley from 1750-2019, CenTrends from 1900-2014, GCM MIROC from 1850-2018, for trend analysis, 5-10 years more or less may not largely influence the final trend, however, if it were a 50-100-year difference, could the trend be biased simply because of the different temporal coverage?

3. GCM simulated precipitation data have poor accuracy compared to temperature. Apparently, in Figure 5, observations (CenTrends) suggested an increasing trend in precipitation over region SS, and four GCM models suggested declining trends, in the end, the synthesized trend was declining. Similar in Figure S1 for EE and S4 for NK. This seems that the synthesized trend is largely influenced by less reliable model simulation instead of the observed trend. The authors may consider justifying this in the discussion section.

4. As mentioned earlier, the results section seems weak, the authors did point out the regional differences of trends in four variables, additional interpretations may need to be added regarding the regional differences, for example, from the perspective of regional climate etc.

5. The authors selected soil moisture because it is a better indicator of crop health than precipitation to study agricultural drought, in conclusion, it's concluded that soil moisture can not be relied upon due to the large uncertainties in both observations and simulations and precipitation should be included given more reliable simulations and observations. This can be confusing and contradictory. As the authors concluded, previous studies using precipitation and this study using soil moisture all detected no consistent trends on droughts. This implies that drought in the study area is not getting worse with increasing temperature and precipitation deficit from the perspective of both meteorology and agriculture. The authors may need to rewrite this properly.

6. The discussion section is hard to read, please consider revising.

Some minor revisions are suggested as follows:

1. Page 2 Line 3, particularly "thorough" or "through" threats to food security?

2. The author did mention that the study period was from the pre-industrial era to 2018 in abstract, but it's hard to tell the study period from the datasets or introduction sections.

3. Some one-sentence paragraphs can be considered to combine with the others based on the logic.

4. Tables 2, 3, and 4 seem outside of the right sections.

5. In section 3.2, more convincing references should be included to justify those assumptions and decisions, for example, the authors assumed that using RefET doesn't influence the overall conclusion, does this mean that the crop coefficients are the same across different regions, if not, PET may vary stronger than RefET.

6. Line 24-28 in the conclusion section should be placed in discussion sections.

7. Units of trends in four variables should be added in Figures 5 and 6 and S1-DS6 in the Supplement figures.

8. Please consider revising some wordy interpretation,  e.g. page 13, We therefore assume for … We therefore..... and the discussion section is so lengthy that readers can easily get lost.

---

## Author Response (AR2)

**Major revision 2 comments**

**Comments from Referees**

The authors have improved the manuscript and the logic is clearer. In general, the interpretation in the result section is still weak. There are still some issues that the authors may consider addressing or clarifying in the manuscript before I can recommend it for publication on ESD.

1. The novelty of this research, as highlighted by the authors, is the usage of multiple methods and models to investigate the impact of precipitation and temperatures on drought trends in EA. According to tables 2 and 3, observational/reanalysis and model data have different spatial resolutions, have they been resampled to the same resolution or used directly in the analysis? Also, the resolution of model data, e.g. GFDL (2.02°\*2.5°), the extent of one region NK (2°N-4.5°N, and 34°E-41°E), it means probably only 6 grids are used to study the trends in variables in this region, whether the global-scale model simulation data is applicable to detect changes over small regions divided in table 1 / Figure 1. For trends derived from datasets with different resolutions, how were they eventually synthesized?

**Response:**

We indeed use all data as it comes with no resampling. The precipitation and temperature data from GFDL thus has a low resolution and indeed the smaller regions are only represented by a few grid cells. As long as the decorrelation length of the drought phenomenon is larger than the grid size of the model, the model can (if validation tests are passed) describe the phenomenon. Compared to other higher resolution models however, the results from low-resolution GCMs do not consistently stand out and also do overlap with observational uncertainty.

Any data set can potentially be used individually to give (albeit unrobust) attribution results. We attempt to synthesize a range of outcomes that can result from individual analyses based on data sets with different properties to arrive at a robust result, i.e., with a representative confidence interval.

*Changes: We added* "resampling or downscaling" *to the sentence* "Note that we use the data as it is available without applying any additional bias correction, resampling or downscaling."

*Furthermore we added* "The results from low resolution GCMs do not consistently stand out compared to higher resolution models and also do overlap with observational uncertainty." to the results section."

2. About datasets in section 2.2, the authors selected 35yrs or longer, multiple datasets spanning different lengths of years were used, e.g. observation Berkeley from 1750-2019, CenTrends from 1900-2014, GCM MIROC from 1850-2018, for trend analysis, 5-10 years more or less may not largely influence the final trend, however, if it were a 50-100-year difference, could the trend be biased simply because of the different temporal coverage?

**Response:**

Note that in Section 3.2 (Assumptions and decisions),

- we explain (point 8) that "Differences in trends can arise due to different time periods and lengths of datasets, which are generally shorter for observations and reanalyses than for model simulations. However, we consider the use of all available observational and reanalysis data and different model framings to lead to a more complete and robust attribution statement."
- we specify that we use Berkeley from 1920 in region SS and from 1900 in all other regions. Thanks to the reviewer's comment we see now that this adjusted start date was not carried over to Table 2.

The model data sets (Table 3) have very similar time spans (except weather@home), where we should realise that most GMST rise occurs in the second half of the 20th century, so starting in 1850 or 1900 does not make a big difference. Some of the observational series e.g. ERA-I (1979-2018) miss the early part of the GMST rise, but due to the necessity of using observations we do not want to exclude these series even though they are shorter. We extrapolate trends from these observational series backwards in time but not without consequence: the extrapolation makes the uncertainty of the trend larger. We added this to assumption 8.

Looking at the 6 regions, ERA-I temperature data does not show a systematic difference: the trend is sometimes larger and sometimes smaller than for the Berkeley data set.

Changes:

The "time period used" for Berkeley Earth given in table 2 has been changed from 1750-2019 to 1900-2018. Also for CRU-TS4 and ERA-I the end date has been changed from 2019 to 2018.

Assumption 8: "Differences in trends can arise due to different time periods and lengths of datasets, which are generally shorter for observations and reanalyses than for model simulations. Extrapolation between the first half of the 20th century and pre-industrial does not make a big difference, as most GMST rise occurs in the second half of the 20th century. For shorter observational series the difference is larger. However, we consider the use of all available observational and reanalysis data and different model framings to lead to a more complete and robust attribution statement."

3. GCM simulated precipitation data have poor accuracy compared to temperature. Apparently, in Figure 5, observations (CenTrends) suggested an increasing trend in precipitation over region SS, and four GCM models suggested declining trends, in the end, the synthesized trend was declining. Similar in Figure S1 for EE and S4 for NK. This seems that the synthesized trend is largely influenced by less reliable model simulation instead of the observed trend. The authors may consider justifying this in the discussion section.

**Response and changes:**

Part of the reviewer's remarks probably concern the best estimate of the trend from CenTrends and GCM models. It is however extremely important not to rely on the best estimate as the uncertainties are large. The CenTrends confidence interval spans zero, with a substantial fraction of the interval on both the negative and positive side, i.e. although the best estimate lies on the positive side, the observational results clearly encompass 'no change'. We would say that only two of the four GCMs the reviewer mentions suggest declining trends whereas the other two GCM results span zero, with one weighted more towards the positive side and the other to the negative. The final synthesized result for region SS is communicated as a negative non-significant trend.

We would furthermore like to emphasize that first all model results are combined (into the dark red bar in the synthesis figures) and all observational results are combined (into the dark blue bar in the synthesis figures). Only afterwards the dark bars are combined (with weighting dependent on the uncertainties of these bars). So all models together and all observations together contribute to the synthesized value with one estimate (including uncertainty estimates).

On this topic we add: "Firstly, we compute the weighted average of **the synthesized values** for models and observations, neglecting model uncertainties beyond the model spread: this is indicated by the magenta bar. ... we also use the more conservative estimate of an unweighted average of the **synthesized values for** observations and models"

We furthermore assume that the reviewer means Figure S1 for WE (not EE). We agree that from the text it is not clear that models and observations do not always fully agree on the trend, although it is clear from the figures.

To clarify this in the text as well, we add to the synthesis results section:

"The more the magenta bar is centered in the white box, the better the models agree with observations and the more we trust our attribution statement"

**And**

"For precipitation, regions WE and NK show a positive but non-significant trend, although in region WE models and observations only partially overlap. In region NS there is a small positive trend, regions EE and CK show no trend (for EE only with partial overlap of models and observations), and region SS a negative, non-significant trend."

**4. As mentioned earlier, the results section seems weak, the authors did point out the regional differences of trends in four variables, additional interpretations may need to be added regarding the regional differences, for example, from the perspective of regional climate etc.**

**Response:**

We acknowledge that an interpretation from a regional climate perspective is missing. The division of the region into smaller subregions was necessary as we only want to study changes over a homogeneous region. The results do not change our motivation for this decision. However, taking all uncertainties into account, the differences between the regions are very small and not clearly related to the regional climate, so we cannot draw conclusions based on the different regional climates.

**Changes:**

We added to the end of the results section that "While it would be desirable to link the overall findings to differences in regional climate, the differences in the synthesized results between regions are too small relative to confidence intervals to be able to say anything meaningful. It was nevertheless necessary to divide the study area into homogeneous regions, so that extremes experienced within each region are representative for that region and inhomogeneity is not influencing the location of the occurrence of extremes."

5. The authors selected soil moisture because it is a better indicator of crop health than precipitation to study agricultural drought, in conclusion, it's concluded that soil moisture can not be relied upon due to the large uncertainties in both observations and simulations and precipitation should be included given more reliable simulations and observations. This can be confusing and contradictory. As the authors concluded, previous studies using precipitation and this study using soil moisture all detected no consistent trends on droughts. This implies that drought in the study area is not getting worse with increasing temperature and precipitation deficit from the perspective of both meteorology and agriculture. The authors may need to rewrite this properly.

Response: Thank you for bringing these sources of potential confusion to our attention.

Soil moisture is indeed more indicative of crop health than precipitation in the study of agricultural drought but we also know that precipitation records are longer and more widespread than soil moisture measurements. If there had been a strong trend in soil moisture our conclusion would have been based on this trend. However, as we see no trend in soil moisture emerging from natural variability, we can not make more robust statement on trends in drought based on soil moisture. After concluding this, we argue that in that case, we can also rely on results obtained from using the longer precipitation records.

Perhaps there is also confusion over the chronological order in which this study developed. Previous trend studies for this part of Africa indeed do not agree on the sign of the trend in precipitation. However, although there is disagreement in reported results, the disagreement lies within observational uncertainty, according to Philip et al., 2018a. We referred to this as detecting no consistent trend on (meteorological) drought. Motivated by reports of a cluster of recent droughts and the request to understand if, despite no evident trend in precipitation, increasing temperatures could be exacerbating drought, we investigated if more insight can be gained by additionally examining the variables PET and soil moisture that are more closely related to crop health than precipitation. We were aware that precipitation measurements are the most reliable, but it is in our opinion still worth investigating if soil moisture and PET show a signal.

**Changes:**

In the introduction we add: "In this study, we aim to understand if, despite no evident trend in precipitation, increasing temperatures could be exacerbating drought."

In the conclusion we change the relevant text to:

"Due to the large uncertainties in both soil moisture observations and simulations, we find no trend emerging from natural variability."

And: "We conc

"We conclude that, although soil moisture is the prefered indicator of agricultural drought, we recommend that any soil moisture analysis be supplemented with precipitation analysis due to the superior reliability of precipitation measurements and the large influence of precipitation on drought in this region. Besides, soil moisture also has a physical lower limit: once the soil is dry it will remain dry. In water-limited regions an analysis of precipitation is thus a helpful addition".

**6. The discussion section is hard to read, please consider revising.**

**Response:**

We revised the discussions section as follows:

- We shortened some paragraphs and deleted subjects that distracted the reader from the main results.
- We added information and moved text from other sections to the discussions section where we or the reviewer thought this was helpful.
- We reordered the discussions section to improve the flow.
- We revised some wordy interpretation.

**Changes:**

For the new discussions section see the revised manuscript.

**Some minor revisions are suggested as follows:**

1. Page 2 Line 3, particularly "thorough" or "through" threats to food security?

**2. The author did mention that the study period was from the pre-industrial era to 2018 in abstract, but it's hard to tell the study period from the datasets or introduction sections.**

*Response:* Thank you for pointing to this, we will add the years 1900 and 2018 to the introduction as well.

Changes:

P3 L18-19. "Assessments will be based on both observations and climate and hydrological model output on the annual time scale, between the years 1900 (to represent the pre-industrial era) and 2018."

**3. Some one-sentence paragraphs can be considered to combine with the others based on the logic.**

*Response:* We have worked throughout the paper to eliminate these, and to sharpen (and simplify) the writing in general.

Changes:

**4. Tables 2, 3, and 4 seem outside of the right sections.**

**Response:**

We assume that the layout of tables, such as these, are adjusted according to the journal's requirements during the typesetting stage.

Changes:

5. In section 3.2, more convincing references should be included to justify those assumptions and decisions, for example, the authors assumed that using RefET doesn't influence the overall conclusion, does this mean that the crop coefficients are the same across different regions, if not, PET may vary stronger than RefET.

Response:

Firstly, we acknowledge that, locally, strong long term trends in land cover could enhance or counteract the reported trends, however this was not the focus of our study. We focus on climate-induced changes rather than land cover-induced changes. Our conclusions are therefore valid for the chosen large study regions, under the condition that there are no strong changes in land use or soil physical conditions in time. Secondly, we remark that in the context of this paper, we don't convert reference ET to PET, nor would we convert reference ET to crop ET using crop coefficients, because it would not be relevant to our research purposes. This study neither needs nor uses crop coefficients as we are only interested in evaporative demand in its purest sense-i.e., as the atmospheric control driving upward moisture flux in the land-atmosphere system. One would use crop coefficients to mediate reference ET towards an estimation of crop evaporation, a value that would then not be a measure of evaporative demand but would instead approach actual evapotranspiration (ET). Even if we were to want to apply crop coefficients to our estimate of reference ET, any crop coefficients we used would be (i) so inaccurate as to be meaningless at the large spatial scales of our analysis, and (ii) different for each of the different metrics of E0 that we use. Further, many hydrologists would start from the perspective of the differences between PET and ET0 being predicated mostly on the surface assumptions involved (open water for PET, a reference crop for reference ET) to argue that assumption #6 in the text mis-states the relationship between PET and reference ET.

A few words on the mix of metrics that we use for evaporative demand (Hamon, Priestley-Taylor, Penman-Monteith). First, even though we have used the abbreviation "PET," we are not actually using PET; instead, we use evaporative demand ( $E_0$ ), which is the name for the concept of (i) the theoretical thirst of the atmosphere, or (ii) the energy limit on evaporation, or (iii) the amount of water that would evaporate were there enough water to meet the need--they're all conceptually the same in the context of this paper. Second,  $E_0$  is an umbrella term that has three specific definitions:

1. potential ET (PET), which is the original, defined by Penman in 1948 as evaporation from an open-water surface or well-watered grass (depending on one's reading of Penman (1948));

2. reference ET  $(ET_0)$ , which is defined as the water evaporating from a specific, well-defined crop surface (the reference crop)--and is what we've used to estimate Eo;

3. and pan evaporation, which is a physical observation of evaporation from the small open-water surface in a pan.

Third, as each of these three  $E_0$  definitions assumes (or observes) evaporation from a surface at a variety of spatial scales, and has a variety of parameterizations (in the case of PET and  $ET_0$ ) or instruments (in the case of pan evaporation) that all make different

assumptions about which drivers are important (temperature; temperature and radiation; and temperature, radiation, wind speed, and humidity in Hamon, Priestley-Taylor, and Penman-Monteith, respectively), it is no surprise that they all yield different values of E0. In the case of PET and ET0, an inexhaustive list of parameterizations includes Thornthwaite, Blaney-Criddle, Hamon, Hargreaves-Samani, Turc, Makkink, Penman, Priestley-Taylor, and Penman-Monteith. Of these, we use Hamon, Priestley-Taylor, and Penman-Monteith. Penman and Blaney-Criddle are described as both PET and ET0, depending on whom you're reading: this is clumsy writing. Penman-Monteith can be both PET and ET0, depending on which parameter values one is using: this is a flexible equation. In the face of all of these uncertainties, the ensemble of models, drivers, and  $E_0$  parameterizations employed in this study is a proven technique for estimating the overall effect of evaporative demand -- in this case, on drought. In fact, this convergence-of-evidence approach is the backbone of operational drought monitoring. To sum up: we do not use crop coefficients nor do we need to; and we should not have stated that we're concluding on PET, but have instead now defined  $E_0$  as above and used the abbreviation "E0" where we previously had "PET" and the term "evaporative demand" where we previously used "potential evaporation" (or "potential evapotranspiration").

**Changes:**

We added the following text in the introduction:

"Ideally, we would study the influence of temperature on soil moisture via ET, however observational records are very limited in time and space and, as the spatial decorrelation lengths of ET are short, their informational value is limited. We therefore analyse evaporative demand ( $E_0$ ); sometimes also referred to as "potential evapotranspiration," or PET, although this is strictly only one metric of  $E_0$ .  $E_0$  is the amount of evaporation that would occur under prevailing meteorological conditions, if an unlimited supply of water were available; in that sense,  $E_0$  measures the thirst of the atmosphere.  $E_0$  is calculable as a function of temperature, humidity, solar radiation, and wind speed. We use a variety of common parameterizations of  $E_0$  that includes both potential evapotranspiration and reference evapotranspiration and that ranges in physical representation and complexity from simple estimates based solely on temperature (the Hamon equation), through estimates that also include solar radiation as a driver (the Priestley-Taylor equation), to ultimately, fully physical estimates that further include humidity and wind speed as drivers (the Penman-Monteith equation). All necessary drivers are available for both observations and model simulations. In this manner, we bracket the complexity in  $E_0$  parameterizations in a convergence-of-evidence approach familiar to the drought-monitoring community.

We investigate  $E_0$  as a means to study the influence of temperature on soil moisture, however, for regions that are irrigated or where irrigation is being considered,  $E_0$  itself can be regarded as more relevant than soil moisture as a measure of drought tendency."

We added an assumption to the list in section 3.2:

"In using our variety of  $E_0$  metrics, we do not convert reference evapotranspiration (such as that drawn from the MERRA-2 dataset (Hobbins et al., 2018) to PET, nor do we use crop coefficients to convert reference evapotranspiration to crop evapotranspiration because doing so would not be relevant to the research purposes. Our study is only interested in evaporative demand in its purest sense - i.e., as the atmospheric control driving upward moisture flux in the land-atmosphere system. In any case, crop coefficients we used would be (i) so inaccurate as to be meaningless at the large spatial scales of our analysis, and (ii) different for each of the different metrics of  $E_0$  that we use. The ensemble of  $E_0$  values generated by our variety of  $E_0$  metrics will ensure that significant trends generated are robust."

And we changed PET into  $E_0$  throughout the paper.

**6. Line 24-28 in the conclusion section should be placed in discussion sections.**

*Response:* We agree that these lines are written in a way that belongs more to the discussion. One of our conclusions is, however, that precipitation should still be considered a good drought indicator in this region, so we will add a sentence to that effect in the conclusions.

*Changes:* We moved lines 24-28 from the conclusions to the discussion. To the conclusions we added "Soil moisture is the prefered indicator of agricultural drought, however we recommend that any soil moisture analysis be supplemented with precipitation analysis due to the superior reliability of precipitation measurements and the large influence of precipitation on drought in this region".

**7. Units of trends in four variables should be added in Figures 5 and 6 and S1-DS6 in the Supplement figures.**

*Response:* trends are in [units of the study variable]/K, so we add for precipitation and PET [mm/day/K], for Temperature [K/K] and for soil moisture [/K].

Changes: we added the units to the paper

8. Please consider revising some wordy interpretation, e.g. page 13, We therefore assume for ... We therefore..... and the discussion section is so lengthy that readers can easily get lost.

*Response:* We agree the text was sometimes too wordy and the discussions section contained information that could easily distract the reader from the main results. We revised wordy interpretation throughout the whole manuscript, and rearranged the discussions section.

Changes: see revised manuscript.

**Impact of precipitation and increasing temperatures on drought trends in **eastern Eastern** Africa**

Sarah F. Kew1, Sjoukje Y. Philip1, Mathias Hauser2, Mike Hobbins3,4, Niko Wanders5, Geert Jan van Oldenborgh6, Karin van der Wiel6, Ted I.E. Veldkamp1, Joyce Kimutai7, Chris Funk8,9, and Friederike E.L. Otto10

1Institute for Environmental Studies, Vrije Universiteit, Amsterdam, The Netherlands
2Institute for Atmospheric and Climate Science, ETH Zurich, Zurich, Switzerland
3Cooperative Institute for Research in Environmental Sciences, University of Colorado Boulder, Boulder, Colorado
4Physical Sciences Laboratory, NOAA/Earth System Research Laboratories, Boulder, Colorado
5Department of Physical Geography, Utrecht University, Utrecht, the Netherlands
6Royal Netherlands Meteorological Institute (KNMI), De Bilt, The Netherlands
7Kenya Meteorological Department, Nairobi, Kenya
8U.S. Geological Survey Center for Earth Resources Observation and Science, Sioux Falls, South Dakota
9University of California, Santa Barbara, Santa Barbara, California
10School of Geography and the Environment, University of Oxford, Oxford, UK

Correspondence: Sarah F. Kew (sarah.kew@knmi.nl)

**Abstract.** In eastern Eastern Africa droughts can cause crop failure and lead to food insecurity. With increasing temperatures, there is an a priori *a priori* a sumption that droughts are becoming more severe, however, However, the link between droughts and climate change is not sufficiently understood. In the current study Here we investigate trends in long-term agricultural drought and the influence of increasing temperatures and precipitation deficits.

- Using a combination of models and observational datasets, we studied trends, spanning the period from 1900 (to represent the approximate pre-industrial eraconditions) to 2018, for six regions in eastern Eastern Africa in four drought-related annually averaged variables — soil moisture, precipitation, temperature and, as a measure of evaporative demand, potential evapotranspiration (PETevaporative demand ( $E_0$ ). In standardized soil moisture data, we found no discernible trends. Precipitation was found to have a stronger The strongest influence on soil moisture variability than temperature or PET was from precipitation,
- 10 especially in the drier, or water-limited, study regions. The : temperature and E0 did not demonstrate strong relations to soil moisture. However, the error margins on precipitation-trend estimates are however large and no clear trend is evident. We find , whereas significant positive trends were observed in local temperatures. However, the influence of these on soil moisture annual trends appears limited. The trends in PET E0 are predominantly positive, but we do not find strong relations between PET E0 and soil moisture trends. Nevertheless, the PET trend E0-trend results can still be of interest for irrigation purposes
   15 because it is PET E0 that determines the maximum evaporation rate.

We conclude that, until now, the impact of increasing local temperatures on agricultural drought in eastern Eastern Africa is limited and we recommend that any soil moisture analysis be supplemented by an analysis of precipitation deficit.

**1 Introduction**

5

25

In eastern Eastern Africa, drought has occurred throughout known history and the phenomenon has incurred with significant impacts on the agricultural sector and the economy, particularly thorough through threats to food security. It is therefore important to examine the role of anthropogenic climate change in drought, particularly in the face of the large-scale droughts of 2010/11, 2014 and 2015 in Ethiopia, and the 2016/17 drought in Somalia, Kenya, and parts of Ethiopia and surrounding countries, which have recently raised the spectre of climate change as a risk multiplier in the region.

Droughts are triggered and maintained by a number of factors and their interactions, including meteorological forcings and variability, soil and vegetation feedbacks, and human factors such as agricultural practices and management choices, including

- 10 irrigation and grazing density (van Loon et al., 2016). Accordingly, there are several definitions of drought in common use (Wilhite and Glantz, 1985): meteorological drought (precipitation deficit), hydrological drought (low streamflow), agricultural drought (low soil moisture) and socioeconomic drought (including water supply and demand). This complexity of droughts poses challenges for their attribution. It is not straightforward to disentangle these interacting factors, but over a long time period long periods it may be possible that a signal can be detected 
[revised manuscript text omitted]
                                     | Full name                             | Time period          | Spatial reso-  | ReferenceCitations(s)              |  |  |
|---------------------------------------------------|---------------------------------------|----------------------|----------------|------------------------------------|--|--|
| dataset                                           |                                       | used                 | lution (°lat x |                                    |  |  |
|                                                   |                                       |                      | °lon)          |                                    |  |  |
| Observatational/re                                | Observatational/reanalysis data set   |                      |                |                                    |  |  |
| CenTrends (prcp)                                  | Centennial Trends data set            | 1900–2014            | 0.1x0.1        | Funk et al. (2015)                 |  |  |
| CRU TS4 (temp)                                    | CRU TS4.01                            |                      | 0.5x0.5        | Harris et al. (2014)               |  |  |
|                                                   |                                       | <del>1901–2019</del> |                |                                    |  |  |
|                                                   |                                       | 1901-2018            |                |                                    |  |  |
| Berkeley (temp)                                   | Berkeley Earth                        | <del>1750_2019</del> | 1.0x1.0        | Rohde et al. (2013b, a)            |  |  |
|                                                   |                                       | 1900-2018            |                |                                    |  |  |
| ERA-I                                             | ERA-Interim                           |                      | 0.5x0.5        | Dee et al. (2011)                  |  |  |
|                                                   |                                       | <del>1979–2019</del> | 0.5.0.0        |                                    |  |  |
|                                                   |                                       | 1979-2018            |                |                                    |  |  |
| Observation-driven hydro/impact model             |                                       |                      |                |                                    |  |  |
| LPJmL-WFDEI                                       | Lund-Potsdam-Jena managed Land -      | 1971–2010            | 0.5 x 0.5      | Bondeau et al. (2007); Rost et al. |  |  |
| (soil moisture)                                   | WATCH-Forcing-Data-ERA-Interim        |                      |                | (2008); Schaphoff et al. (2013);   |  |  |
|                                                   |                                       |                      |                | Weedon et al. (2014)               |  |  |
| PCRGLOB-                                          | PCRaster GLOBal Water Balance         | 1971–2010            | 0.5 x 0.5      | Sutanudjaja et al. (2018); Weedon  |  |  |
| WFDEI (soil                                       | model - WATCH-Forcing-Data-ERA-       |                      |                | et al. (2014)                      |  |  |
| moisture)                                         | Interim                               |                      |                |                                    |  |  |
| CLM-ERA-I (soil                                   | Community Land Model version 4 -      | 1979–2016            | 0.5 x 0.5      | Oleson et al. (2010)               |  |  |
| moisture, $\frac{\text{PETE}_0}{\text{PETE}_0}$ ) | ERA-Interim                           |                      |                |                                    |  |  |
| CLM-WFDEI                                         | Community Land Model version 4 -      | 1979–2013            | 0.5 x 0.5      | Lawrence et al. (2011); Weedon     |  |  |
| (soil moisture,                                   | WATCH-Forcing-Data-ERA-Interim        |                      |                | et al. (2014)                      |  |  |
| PETE 0 )                               |                                       |                      |                |                                    |  |  |
| FLDAS (soil                                       | Famine Early Warning Systems Net-     | 1981–2018            | 0.1 x 0.1      | McNally et al. (2017)              |  |  |
| moisture)                                         | work (FEWS NET) Land Data Assimi-     |                      |                |                                    |  |  |
|                                                   | lation System                         |                      |                |                                    |  |  |
| MERRA Ref-ET                                      | Modern-Era Retrospective analysis for | 1980–2018            | 0.125 x 0.125  | Hobbins et al. (2018)              |  |  |
| (PETE 0 )                              | Research and Applications Reference   |                      |                |                                    |  |  |
|                                                   | Evapotranspiration                    |                      |                |                                    |  |  |

Table 3. Model data used in this study.

| Model dataset                              | Full name                           | Time period    | Spatial reso-  | ReferenceCitations(s)                |  |  |  |
|--------------------------------------------|-------------------------------------|----------------|----------------|--------------------------------------|--|--|--|
|                                            |                                     | used           | lution (°lat x |                                      |  |  |  |
|                                            |                                     |                | °lon)          |                                      |  |  |  |
| GCM/RCM                                    | GCM/RCM                             |                |                |                                      |  |  |  |
| GFDL                                       | GFDL-ESM2M, Geophysical Fluid       | 1861–2018      | 2.02x2.5       | Dunne et al. (2012, 2013)            |  |  |  |
|                                            | Dynamics Laboratory - Earth System  |                |                |                                      |  |  |  |
|                                            | Model 2M                            |                |                |                                      |  |  |  |
| HadGEM                                     | HadGEM2-ES, Hadley Centre Global    | 1859–2018      | 1.25x1.88      | Collins et al. (2011); Jones et al.  |  |  |  |
|                                            | Environmental Model version 2-ES    |                |                | (2011)                               |  |  |  |
| IPSL                                       | IPSL-CM5A-LR, Institut Pierre Simon | 1850–2018      | 1.89x3.75      | Dufresne et al. (2013)               |  |  |  |
|                                            | Laplace - CM5A-LR                   |                |                |                                      |  |  |  |
| MIROC                                      | MIROC5, Model for Interdisciplinary | 1850–2018      | 1.4x1.4        | Watanabe et al. (2010)               |  |  |  |
|                                            | Research on Climate - version 5     |                |                |                                      |  |  |  |
| EC-Earth                                   | EC-Earth 2.3                        | 1850–2018      | 1.12x1.125     | Hazeleger et al. (2012)              |  |  |  |
| w@h (temp, prcp,                           | Weather@home                        | 2005–2016 and  | 0.11x0.11      | Massey et al. (2015); Guillod et al. |  |  |  |
| soil moisture)                             |                                     | counterfactual |                | (2017)                               |  |  |  |
|                                            |                                     | climate        |                |                                      |  |  |  |
| Hydro/impact models                        |                                     |                |                |                                      |  |  |  |
| H08 (soil mois-                            | H08                                 | 1861–2018      | 0.5x0.5        | Hanasaki et al. (2008a, b)           |  |  |  |
| ture, $\frac{\text{PETE}_0}{\text{E}_0}$ ) |                                     |                |                |                                      |  |  |  |
| LPJmL (soil                                | Lund-Potsdam-Jena managed Land      | 1861–2018      | 0.5x0.5        | Bondeau et al. (2007); Rost et al.   |  |  |  |
| moisture, PETE 0 )       | model                               |                |                | (2008); Schaphoff et al. (2013)      |  |  |  |
| PCRGLOB (soil                              | PCRGLOB-WB, PCRaster GLOBal         | 1861–2018      | 0.5x0.5        | Sutanudjaja et al. (2018)            |  |  |  |
| moisture, $\frac{\text{PETE}_0}{1}$ )      | Water Balance model                 |                |                |                                      |  |  |  |
| WaterGAP2 (soil                            | Water Global Analysis and Progress  | 1861–2018      | 0.5x0.5        | Müller Schmied et al. (2016)         |  |  |  |
| moisture, PETE 0 )       | Model version 2                     |                |                |                                      |  |  |  |

---

## Author Response (AR3)

**Reply to reviewer points and further edits**

**(1) Figure 1, the right-hand sub-figure is blurry with almost all text characters unclear.**

The figure comes from Pricope et al. 2013:

Pricope, N. G., Husak, G., Lopez-Carr, D., Funk, C., and Michaelsen, J.: The climate-population nexus in the East African Horn: Emerging degradation trends in rangeland and pastoral livelihood zones, Global Environmental Change, 23, 1525 – 1541, https://doi.org/https://doi.org/10.1016/j.gloenvcha.2013.10.002, 2013.

Before submitting the manuscript, I asked the author for a higher quality image and that is the one I have used. It looks no better in her original paper (see screen shot below). Therefore I have left it as it was.

[Figure]

**Fig. 1.** (a) FEWS Net livelihoods zones map for Ethiopia, Kenya, and Somalia; (b) population density change map from 1990 to 2010 (based on the calculation in Eq. (1)).

woody savannas (often associated with shrub encroachment due to overgrazing, drying or changing fire regimes). However, this aggregation does not affect the results of this work as our main

the literature use this data for regional-scale analyses of land cover change. Due to the regional scale of our analysis, the lack of appropriate resolution ground validation data, and known issues

**(2) I will also suggest unifying the font characteristics (e.g., the font weight is bold or not) between Figure 2 and Figure 3. Some text characters (e.g., the labels of X-axis and Y-axis as well as the legend) could be enlarged to make them clearer in both figures.**

While I appreciate the reviewer's eye for detail, the model experiments and production of the figures was a team effort and these 2 figures come from different sources. I would certainly change it if Figure 3 had vector graphics, but this is not the case. As it is not essential for the understanding of the paper, this suggestion has not been taken on. I have changed the font size for the x-axis and y-axis and legend in Figure 2, and Figure 3 has been enlarged.

**(3) Figure 5, the font size of lable in Y-axis label in the sub-figure describing PET could be enlarged to make them unified to other sub-figures.**

The individual panels have now been scaled by the same scaling factor so that the sub-figures now have the same font sizes.

**Further edits**

References to Philip et al. 2019 (now Philip et al. 2020, published) and van Oldenborgh et al. 2019 (now van Oldenborgh et al. 2020, under review for a different journal) have been changed.

In Figure 4 the incorrect notation for the region West Ethiopia has been changed from WK to WE, and units used for precipitation and temperature are now mentioned.

The layout of the synthesis figure plots (Figure 5 and supplement figures) has been changed to be more space efficient, when combining the subpanels to one plot.

The way that panels are referred to in the figure captions has been standardised.

On page 4 L19-20, "tables 2 and ??" has been corrected to "tables 2 and 3".

*These edits do not change the meaning of the content of the text or figures.*